# Anti-human-TIGIT agonistic antibody ameliorates autoimmune diseases by inhibiting Tfh and Tph cells and enhancing Treg cells

Marenori Kojima [1], Katsuya Suzuki[1], Masaru Takeshita[1], Masaki Ohyagi[2], Mana Iizuka[2], Humitsugu Yamane[1], Keiko Koga[3], Taku Kouro[3], Yoshiaki Kassai[3], Tomoki Yoshihara[4], Ryutaro Adachi[4], Kentarou Hashikami[4,5], Yuichiro Ota[1], Keiko Yoshimoto[1], Yuko Kaneko[1], Rimpei Morita[2,6], Akihiko Yoshimura[2] & Tsutomu Takeuchi [1]✉

T cells play important roles in autoimmune diseases, but it remains unclear how to optimally manipulate them. We focused on the T cell immunoreceptor with Ig and ITIM domains (TIGIT), a coinhibitory molecule that regulates and is expressed in T cells. In autoimmune diseases, the association between TIGIT-expressing cells and pathogenesis and the function of human-TIGIT (hu-TIGIT) signalling modification have not been fully elucidated. Here we generated anti-hu-TIGIT agonistic monoclonal antibodies (mAbs) and generated hu-*TIGIT* knock-in mice to accurately evaluate the efficacy of mAb function. Our mAb suppressed the activation of CD4$^+$ T cells, especially follicular helper T and peripheral helper T cells that highly expressed TIGIT, and enhanced the suppressive function of naïve regulatory T cells. These results indicate that our mAb has advantages in restoring the imbalance of T cells that are activated in autoimmune diseases and suggest potential clinical applications for anti-hu-TIGIT agonistic mAbs as therapeutic agents.

[1] Division of Rheumatology, Department of Internal Medicine, Keio University School of Medicine, Shinjuku-ku, Tokyo, Japan. [2] Department of Microbiology and Immunology, Keio University School of Medicine, Shinjuku-ku, Tokyo, Japan. [3] Immunology Unit, Pharmaceutical Research Division, Takeda Pharmaceutical Company Limited, Fujisawa City, Kanagawa, Japan. [4] Biomolecular Research Laboratories, Pharmaceutical Research Division, Takeda Pharmaceutical Company Limited, Fujisawa City, Kanagawa, Japan. [5] Present address: Axcelead Drug Discovery Partners, Inc., Fujisawa City, Kanagawa, Japan. [6] Present address: Department of Microbiology and Immunology, Nippon Medical School, Bunkyo-ku, Tokyo, Japan. ✉email: tsutake@z5.keio.jp

Treatment of systemic autoimmune diseases has improved with the advent of molecular targeted therapies[1–3], but some patients still cannot control disease activity. The autoantibodies are characteristic of some autoimmune diseases, which are produced by the cooperation of self-reactive T cells and B cells. Follicular helper T (Tfh) and peripheral helper T (Tph) cells were reported to assist B cells[4,5], and they are associated with autoimmune disease pathogenesis[6–9]. Because activation and inactivation of T cells are strictly regulated by signalling mediated by T cell receptors and costimulatory/inhibitory molecules[10], we hypothesised that coinhibitory molecules might be a therapeutic target.

T cell immunoreceptor with Ig and immunoreceptor tyrosine-based inhibitory motif domains (TIGIT) is a unique coinhibitory receptor expressed on effector T, memory T, regulatory T (Treg), and natural killer (NK) cells. TIGIT binds to two ligands, CD112 (PVRL2, nectin-2) and CD155 (poliovirus receptor, PVR), which are expressed on antigen-presenting cells, fibroblasts, endothelial cells, and some cancer cells[11–14]. TIGIT signalling inhibits non-Treg and NK cell activation[12,15]. TIGIT-deficient or TIGIT-inhibited mice are known to exhibit exacerbated symptoms of experimental autoimmune encephalomyelitis (EAE) and collagen-induced arthritis (CIA)[16,17], and conversely, anti-mouse-TIGIT agonistic monoclonal antibodies (mAbs) have been reported to improve EAE symptoms by inhibiting T cell activation[18]. TIGIT signalling is also known to have the potential to enhance the suppressive function of Treg cells[19,20]. Agonistic mAbs to human-TIGIT (hu-TIGIT) were reported to induce to Treg cell effector molecule fibrinogen-like protein 2[19], but there are no reports directly demonstrating the in vivo effect of anti-hu-TIGIT agonistic mAb or analysing its functions.

In this study, we developed anti-hu-TIGIT agonistic mAbs and hu-*TIGIT* knock-in (KI) mice and explored how our mAb acts on a mouse model and human cells in detail. Our findings indicate that anti-hu-TIGIT agonistic mAb can manipulate T cell imbalance, and it has a potential clinical application as therapeutic agents for autoimmune diseases.

## Results

### Correlation between CD4[+] T cell subsets and disease activity in patients with systemic autoimmune diseases.

To confirm in detail whether TIGIT-expressing cells are involved in the pathogenesis of autoimmune diseases, we first checked TIGIT expression in T cells in patients with rheumatoid arthritis (RA), systemic lupus erythematosus (SLE) and Sjögren's syndrome (SjS), where TIGIT expression is known to be changed in peripheral T cells[21–23]. First, we compared the proportions of various T cell subsets and the expression of TIGIT and programmed cell death-1 (PD-1), known as a coinhibitory molecule[24], on each cell subset in those diseases by performing immunophenotyping of peripheral blood (PB) by flow cytometry. Specifically, these features were compared among 10 patients with untreated active RA, 10 patients with active SLE, 20 patients with untreated SjS and 15 healthy controls (HCs) (Supplementary Table 1). Generally, in some patient groups versus the HCs, the proportions of Tfh and Tph cells were significantly higher in the CD4[+] T cell population, while the proportion of CD45RA[+] effector memory T (Temra) cells was significantly higher in the CD8[+] T cell population (Supplementary Fig. 1). Within the CD4[+] T cell compartment, many TIGIT-expressing cells were observed in memory subsets, especially in Tfh (defined as CD45RA[-] CXCR5[+]) and Tph cells (defined as CXCR5[-] PD-1[high]), whereas non-Tfh/Tph cells showed a lower proportion of TIGIT than the other subsets. Conversely, high levels of PD-1-expresing cells were observed in all memory subsets, with no difference between Tfh and non-Tfh/ Tph cells. Compared with the HCs, there were significantly more TIGIT- and PD-1-expressing cells from memory subsets in RA, SLE, and SjS groups (Fig. 1a, b). In CD8[+] T cells, high levels of TIGIT- and PD-1-expressing cells were also observed in the memory subsets, but unlike CD4[+] T cells, CD8[+] T cells showed no differences between the HCs and patients with respect to TIGIT and PD-1 expression levels (Fig. 1c, d).

Next, to explore whether these differences in inhibitory receptor-expressing T cell subsets were related to autoimmune disease activities, we analysed the correlations between the proportions of TIGIT- and PD-1-expressing cells and disease activity indexes. In the RA patients, TIGIT expression on effector memory T (Tem), Tfh and Tph cells was positively correlated with the disease activity score-C-reactive protein (DAS28-CRP) ($P = 0.68$, 0.70 and 0.67, respectively). In the SLE patients, TIGIT expression on Tem and Tph cells was positively correlated with the SLE disease activity index ($P = 0.89$ and 0.80, respectively). In the SjS patients, on the other hand, the expression of TIGIT was not correlated with the European League Against Rheumatism (EULAR) SjS disease activity index for any of the subsets tested. Interestingly, non-Tfh/Tph cells were not significantly correlated with disease activity in each autoimmune disease (Fig. 1e). Furthermore, there was no correlation between the expression of PD-1 and disease activity in any of the patient groups (Fig. 1f). These results indicate that TIGIT-expressing cells rather than PD-1-expressing cells are strongly involved in the pathogenesis of RA and SLE.

### Characterisation of developed anti-hu-TIGIT agonistic mAbs and hu-*TIGIT* KI mice.

We hypothesized that inhibitory signals against TIGIT-expressing cells, which are increased in autoimmune diseases, would be useful for treating these diseases. Hence, we developed anti-hu-TIGIT agonistic mAbs in mice. Briefly, recombinant hu-TIGIT-mFc-His protein was injected into CD2F1 mice to obtain anti-hu-TIGIT mAbs. We then screened the hybridomas and selected the seven mAbs, which were tested for agonistic activity, antagonistic activity, and cytotoxicity by measuring the luminescence values using luciferase reporter cells (Supplementary Fig. 2a). The seven mAbs possessed agonistic activity but lacked antagonistic activity and cytotoxicity (Supplementary Fig. 2b). Next, to identify the most active mAb, we compared the inhibitory activity of the mAbs toward human Tfh cells, already known to have high TIGIT expression (Fig. 1a), in a suppression assay. We sorted the Tfh cells from the HCs and labelled them with CellTrace violet (CTV). Then, we cultured the cells in plates coated with 2 μg ml[−1] anti-human CD3, 1 μg ml[−1] CD28 antibody and 10 μg ml[−1] anti-hu-TIGIT agonistic mAb (or an isotype control) for 4 days and then examined the cultures for cell proliferation by monitoring CTV dilution by flow cytometry. Then, M1-8, which showed the most inhibited Tfh cell suppression, was selected and applied in subsequent experiments (Fig. 2a). We also evaluated its cross-recognition, and M1-8 recognise hu- and macaque-TIGIT, but not mo-TIGIT (Supplementary Fig. 2c).

To directly assess the behaviour of the hu-TIGIT agonistic mAb in mice, we generated hu-*TIGIT* KI mice by inserting hu-*TIGIT* into C57BL/6JJcl-*Tigit* mice (Fig. 2b). We evaluated the immunophenotyping of thymocytes and splenocytes in these mice. T cell development and the weight of spleen in KI mice did not change compared with WT mice, and the expression of Tfh and Tph cells in splenocytes were lower than those in WT mice (Supplementary Fig. 3a, b). We also evaluated the expression patterns of TIGIT in these mice and found that the Tfh cells in the spleens of KI homozygous or heterozygous mice and wild-type (WT) mice had expression patterns that coincided with the respective mouse genotypes (Supplementary Fig. 4a, b), indicating successful generation of hu-*TIGIT* KI mice. TIGIT was expressed

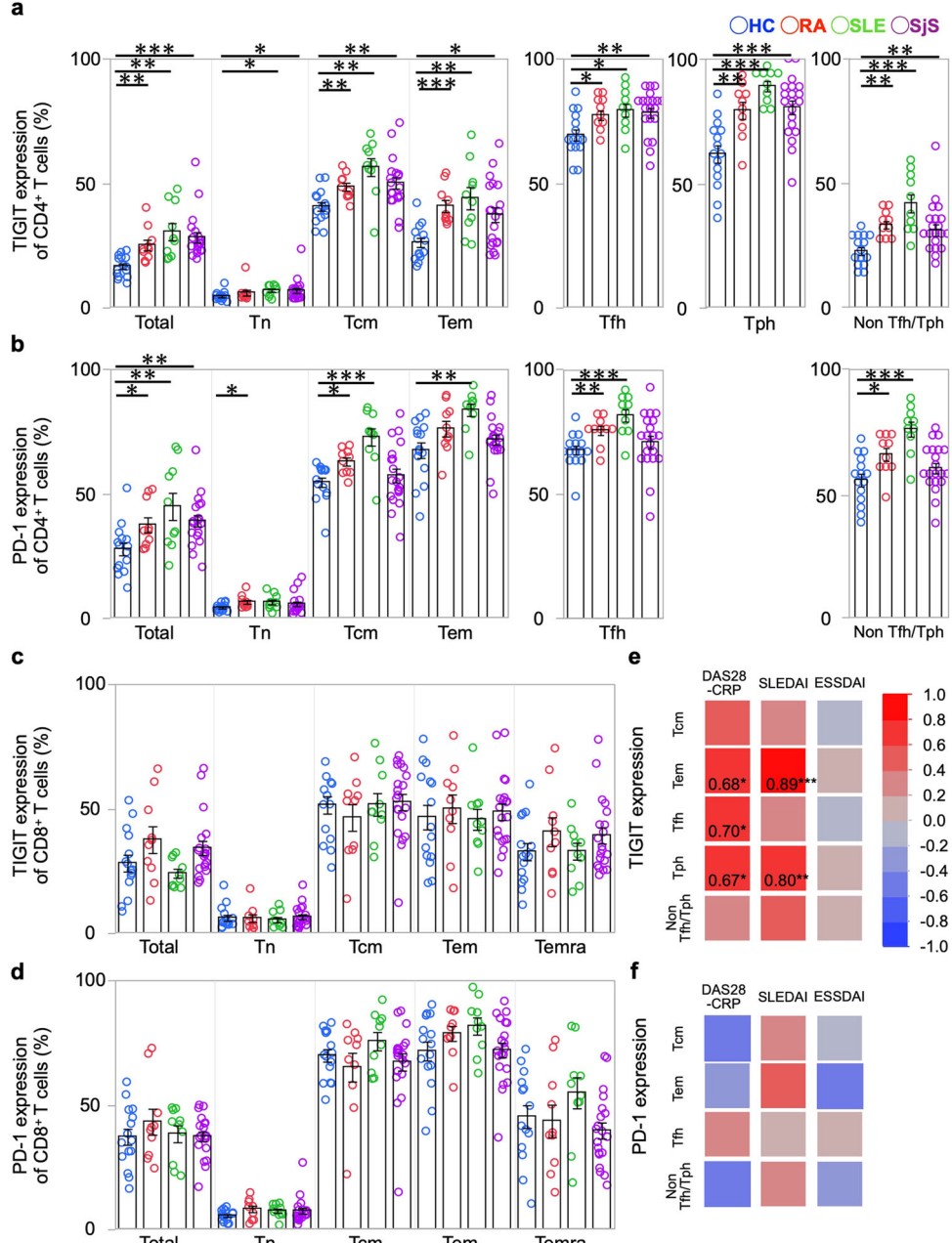

**Fig. 1 TIGIT-expressing CD4+ T cell subsets correlate disease activity in patients with systemic autoimmune diseases.** PBMCs from ten untreated RA patients, ten exacerbated SLE patients, ten untreated SjS patients, and 15 HCs were immunophenotyped. The proportions of TIGIT (**a**) and PD-1 (**b**) at each developmental stage (Tn, Tcm, Tem, Tfh, Tph, non-Tfh/Tph) among CD4+ T cells from the HCs (blue) and the patients with RA (red), SLE (green), and SjS (purple) are shown. The proportions of TIGIT (**c**) and PD-1 (**d**) at each developmental stage (Tn, Tcm, Tem, Temra) among CD8+ T cells from the HCs (blue) and the patients with RA (red), SLE (green) and SjS (purple) are shown. Error bars represent the mean ± SEM, and P values were determined by the Wilcoxon rank sum test. *P < 0.05, **P < 0.01, ***P < 0.001. A heatmap indicating the correlations between the proportions of TIGIT− (**e**) or PD-1- (**f**) expressing cells and disease activity (RA: DAS-28CRP, SLE: SLEDAI, SjS: ESSDAI) is shown. The numbers indicate the correlation coefficients in of Spearman's test. *P < 0.05, **P < 0.01, ***P < 0.001.

in Tfh cells and Treg cells among mouse splenocytes, although the proportions of TIGIT-expressing cells in both KI and WT mouse T cells were lower than those in human cells (Fig. 1a and Supplementary Fig. 4b).

To confirm that an anti-hu-TIGIT agonistic mAb works in KI mice, we utilized an EAE mouse model in which CD4+ T cells contribute to pathogenesis[25] and the clinical scores in WT mice can be suppressed by treatment with a TIGIT agonist[18]. The KI mice were immunized with myelin oligodendrocyte (MOG) peptide and treated with 100 µg mouse−1 anti-hu-TIGIT agonistic

mAb (or an isotype control) on days 0, 2, 4, 10, and 17 (Fig. 2c). The mAb significantly improved the clinical scores (Fig. 2d), confirming that our mAb targeting hu-TIGIT was effective in the EAE model established with hu-*TIGIT* KI mice.

**Anti-hu-TIGIT agonistic mAb suppresses autoantibody production via Tfh cell suppression in an imiquimod-induced lupus model in KI mice.** Since TIGIT-expressing cells were abundant among both human and mouse Tfh cells (Fig. 1 and Supplementary Fig. 4b), we hypothesized that our agonistic mAb

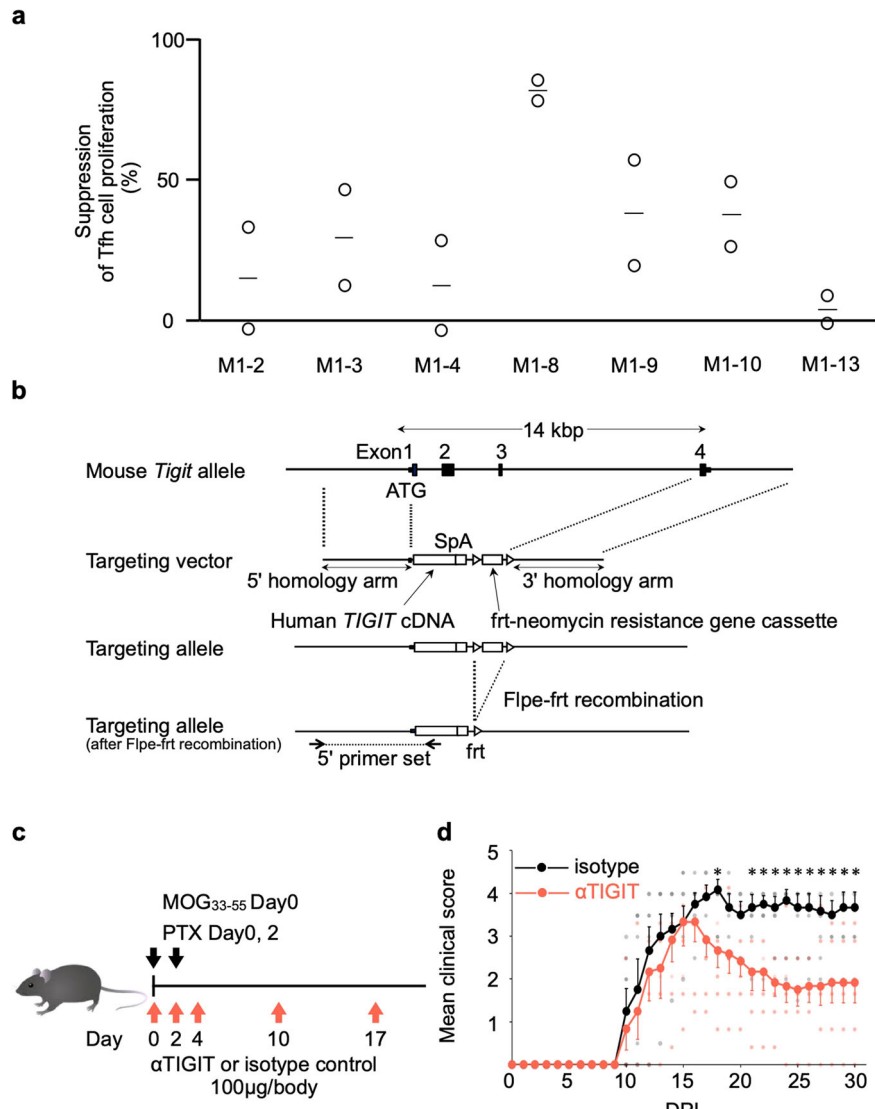

**Fig. 2 Anti-human-TIGIT (anti-hu-TIGIT) agonistic mAb (αTIGIT) ameliorates EAE hu-*TIGIT* knock-in (KI) mice. a** The sorted Tfh cells were labelled with CellTrace violet (CTV) and activated with anti-CD3 and anti-CD28 Abs with plate-bound αTIGIT (or the appropriate isotype control) for 4 days. CTV dilution was analysed by flow cytometry. The suppression rate was calculated as follows: $(1 - (\text{proliferation rate of mAb})/(\text{proliferation rate of isotype})) \times 100$ (mean + SEM) is shown ($n = 2$). Horizontal lines represent the mean of each group. **b** The strategy for generating hu-*TIGIT* KI mice is shown. **c** EAE immunisation and αTIGIT treatment protocols of 12-week-old male hu-*TIGIT* KI mice are shown. **d** EAE mice were divided into treated ($n = 6$, red) and nontreated ($n = 6$, black) groups, and the mean clinical score + or − SEM is shown. $P$ values were determined by a two-tailed Student's $t$ test. *$P < 0.05$.

would be more effective against Tfh cells. As Tfh cells, which facilitate B cell differentiation in the germinal centre (GC) and subsequent antibody production[4], have been reported to be involved in imiquimod (IMQ)-induced lupus pathogenesis in mice[26,27], we evaluated the effect of our mAb in this model. We administered 50 mg kg$^{-1}$ IMQ three times per week and 300 μg mouse$^{-1}$ anti-hu-TIGIT agonistic mAb (or an isotype control) twice per week for 42 days (Fig. 3a). As shown in Fig. 3b, our mAb significantly suppressed splenomegaly and the increase in the number of splenocytes induced by IMQ treatment. To confirm the onset of disease, we evaluated the group administered the isotype without IMQ treatment. Among the splenocytes, the proportion of hu-TIGIT-expressing cells in CD4$^+$ Tem, Tfh, Tph, and Foxp3$^+$ Treg cells was increased by IMQ (Fig. 3c). In IMQ-induced lupus model, the proportion of hu-TIGIT-expressing cells in KI mice was comparable to that of mo-TIGIT-expressing cells in WT mice, and mo-TIGIT-expressing cells in KI mice were not increased

(Supplementary Fig. 5a, b). Our mAb significantly suppressed the proportions of Tem, Tfh (defined as CXCR5$^+$ PD-1$^+$), and Tph (defined as CXCR5$^-$ PD-1$^{high}$)[28] cells in CD4$^+$ T cells (Fig. 3d, e). Conversely, our mAb did not affect the proportion of Foxp3$^+$ Treg cells among CD4$^+$ T cells (Fig. 3f), despite an increase in TIGIT expression (Fig. 3c). Foxp3 expression on Treg cells has been reported to decline under inflammatory conditions, and these Treg cells have been shown to lose their function[29]. Therefore, we checked the proportion of Foxp3$^{high}$ Treg cells. The proportion of Foxp3$^{high}$ Treg cells, which was downregulated by IMQ, among total Treg cells was significantly higher in the mAb+IMQ group than in the isotype+IMQ group (Fig. 3g). Although the mAb had no apparent effect on the overall proportion of B cells with low TIGIT expression (Fig. 3c, h), it did significantly decrease the proportions of GC B cells (defined as CD95$^+$ GL7$^+$) and plasma cells (defined as CD19$^-$ CD138$^+$) (Fig. 3i, j). Similar results for the suppression of Tfh, Tph, GC B and plasma cells were observed in

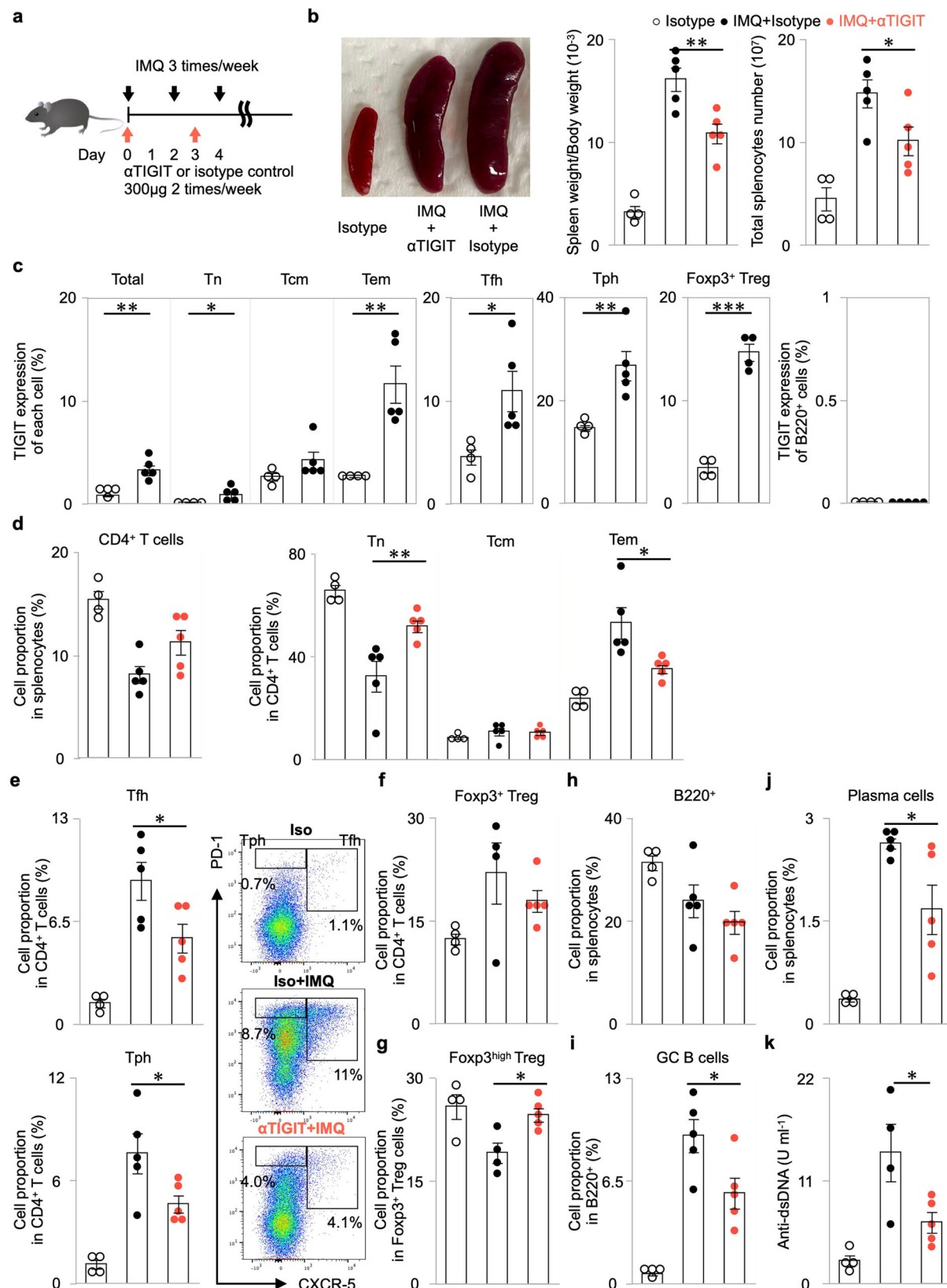

the lymph nodes (Supplementary Fig. 6). Importantly, the elevation of anti-dsDNA antibody, an autoantibody already known to be elevated by IMQ[26,27], was suppressed by the mAb (Fig. 3k). These results clearly showed that our mAb has inhibitory effects on activated B cells by suppressing Tfh and Tph cells and retains Foxp3^high Treg expression in Treg cells.

**A unique Tfh cell-specific increase in TIGIT expression occurs upon activation in vitro.** The IMQ study showed that our mAb worked against activated CD4+ T cells with elevated TIGIT expression. In humans, we first examined how TIGIT expression changes after T cell stimulation. We focused on Tfh cells, which are the highest TIGIT-expressing cells among CD4+ T cells in

**Fig. 3 Anti-human-TIGIT (anti-hu-TIGIT) agonistic mAb (αTIGIT) suppresses T cell-dependent autoantibody production in a hu-*TIGIT* knock-in (KI) imiquimod (IMQ)-induced lupus model in mice. a** The method used to establish IMQ-induced lupus model mice and the αTIGIT treatment protocol of 10-week-old female hu-*TIGIT* KI mice are shown. **b** Hu-*TIGIT* KI mice were divided into three groups: isotype-treated ($n = 4$, black open circle), IMQ- and αTIGIT-treated ($n = 5$, red closed circle), and IMQ- and isotype-treated ($n = 5$, black closed circle) mice. Spleen weight/body weight and total splenocyte number are shown. **c** TIGIT expression of various CD4$^+$ T cell subsets and that of B220$^+$ cells are shown. **d** The proportion of CD4$^+$ T cells in splenocytes and Tn, Tcm, and Tem in CD4$^+$ T cells are shown. **e** Representative flow cytometry plots for Tph cells and Tfh cells and the proportions of those cells are shown. **f** The proportion of Foxp3$^+$ Treg cells among CD4$^+$ T cells and Foxp3$^{high}$ Treg cells (**g**) among Foxp3$^+$ Treg cells are shown (one sample could not be collected in IMQ- and isotype-treated). The proportions of B220$^+$ cells (**h**) in the spleen, germinal centre (GC) B cells (**i**) among B220$^+$ cells and plasma cells (**j**) in total splenocytes are shown. **k** Anti-dsDNA Abs in plasma at 6 weeks are shown (one sample could not be collected in IMQ- and isotype-treated). Error bars represent the mean ± SEM, and P values were determined using a two-tailed Student's t test. *$P < 0.05$, **$P < 0.01$, ***$P < 0.001$.

humans (Fig. 1a) and exhibited inhibited proliferation and function in the IMQ study (Fig. 3). CD4$^+$ T cells from the HCs were divided into Tfh, non-Tfh, and naive T (Tn) cells by CD45RA and CXCR5 (Fig. 4a); in these three subsets, the proportion of both TIGIT and PD-1 were highest in the Tfh cells followed by the non-Tfh cells and then the Tn cells (Fig. 4b).

We sorted the T cells from the HCs into these different TIGIT-expression subsets, stimulated them with equal amounts of anti-CD3/28 antibody-conjugated beads for 4 days and examined the changes in TIGIT and PD-1 expression by flow cytometry. TIGIT$^+$ Tfh cells showed a remarkable increase from 50.9 to 85.2%, whereas TIGIT$^+$ non-Tfh and Tn cells did not show a marked increase (from 27.2 to 35.2% and 2.9 to 2.2%, respectively) (Fig. 4c). The proportion of PD-1, however, increased remarkably in all three groups (Fig. 4d). Mean fluorescence intensity (MFI) also showed that TIGIT was strongly expressed in Tfh cells, while PD-1 was upregulated in all cells (Fig. 4e). These results indicate that a change in TIGIT expression after stimulation is the most pronounced for Tfh cells among these three groups.

**Anti-hu-TIGIT agonistic mAb functionally suppressed Tfh cells in vitro.** Next, to clarify the effect of mAb on TIGIT elevated by T cell stimulation, we investigated the effect of the mAb on human Tfh and non-Tfh cells. We sorted each cell from the HCs and labelled them with CTV. Then, each cell was cultured in plates coated with 2 µg ml$^{-1}$ anti-human CD3, 1 µg ml$^{-1}$ CD28 antibody and 10 µg ml$^{-1}$ anti-hu-TIGIT agonistic mAb (or an isotype control) for 4 days, and the cultures were examined for cell proliferation by monitoring CTV dilution by flow cytometry (Fig. 5a, b). There was no difference in cell viability (Fig. 5a, c), and although both cells were inhibited cell proliferation, a stronger inhibitory effect on cell proliferation was observed for the Tfh cells than the non-Tfh cells (Fig. 5b, d). These results showed that the mAb more strongly inhibited cells with elevated TIGIT expression.

We also investigated the effect of the mAb on the B cell help function of Tfh cells. We cocultured equal amounts of Tfh and CD27$^+$ memory B cells with 0.2 µg ml$^{-1}$ staphylococcal enterotoxin B (SEB) in 10 µg ml$^{-1}$ anti-hu-TIGIT agonistic mAb (or an isotype control)-coated plates for 7 days and evaluated B cell differentiation and IgG production. As expected, compared with the isotype group, the mAb inhibited the differentiation of memory B cells into plasma cells and eventually suppressed antibody production (Fig. 5e–g). These results suggest that our mAb induces similar effects in vitro and in vivo in terms of their inhibitory effects on Tfh cell proliferation and the B cell help function of Tfh cells.

**Anti-hu-TIGIT agonistic mAb functionally activated naive Treg cells in vitro.** Our in vivo results showed that the proportion of Foxp3$^{high}$ Treg cells, which were downregulated by IMQ,

was high in the mAb group, and TIGIT signalling in Treg cells has the potential to enhance suppressive function[19]. Then, we evaluated whether our mAb affects human Treg cells, which can be clearly divided by function, unlike mouse Treg cells. First, we divided naive Treg (FrI) and effector Treg (FrII) cells[30] from CD4$^+$ T cells (Fig. 6a) and examined how the expression of TIGIT and PD-1, which are activated Treg cell markers[19,31], changed after stimulation. As shown in Fig. 6b, each sorted CTV-labelled Treg cell from the HCs was cultured in plates coated with 2 µg ml$^{-1}$ anti-human CD3 and 1 µg ml$^{-1}$ CD28 antibody for 6 h and then cocultured with irradiated PB mononuclear cells (PBMCs) from the same HCs in plates coated with 2 µg ml$^{-1}$ anti-human CD3 antibody for 4 days. The proportions of TIGIT$^+$ and PD-1$^+$ FrI cells were markedly elevated by stimulation, but those of FrII cells were not different, and each mean fluorescence intensity was also similar results (Fig. 6c, d). Importantly, TIGIT, but not PD-1, was expressed in almost all FrI cells after stimulation and FrII cells. These results may indicate that both types of Treg cells are good targets for our agonistic mAb. Next, to clarify TIGIT and PD-1 expression of each Treg cell in autoimmune diseases, we compared them with the patients with active RA and the HCs (Supplementary Table 2). Interestingly, all of them exhibited higher proportions and expression levels in the patients with active RA than in the HCs, and TIGIT$^+$ FrI cells were the most elevated subset (Supplementary Fig. 7).

Finally, we directly evaluated the effect of the mAb on the suppressive function of Treg cells. As shown in Fig. 6e, we cultured the sorted Treg cells in plates coated with 2 µg ml$^{-1}$ anti-human CD3, 1 µg ml$^{-1}$ CD28 antibody and 10 µg ml$^{-1}$ anti-hu-TIGIT agonistic mAb (or an isotype control) for 6 h and then cultured them with irradiated PBMCs from the same HCs in plates coated with 2 µg ml$^{-1}$ anti-human CD3 antibody and cocultured them with equal amounts of CTV-labelled CD25$^-$ responder T cells for 4 days. Responder T cells cocultured with FrI or FrII cells pretreated with the isotype showed a lower proliferation rate than responder T cells alone, and pretreatment of FrI or FrII cells with the mAb enhanced their suppressive function. Both Treg cells exhibited enhanced suppressive function with the mAb, especially FrI cells, which exhibited significantly enhanced suppressive function by approximately twofold (Fig. 6f, g). These study results suggest that the anti-hu-TIGIT agonistic mAb enhanced their suppressive function, especially that of FrI cells with elevated TIGIT expression.

## Discussion
In this study, to develop better therapeutic agents, we focused on manipulating T cells and created anti-hu-TIGIT agonistic mAbs. Our mAb had the certain effect on two mouse models, EAE and IMQ-induced model in vivo. Our mAb demonstrated the potential to control T cell imbalance via simultaneously the suppression of Tfh cells and activation of Treg cells in autoimmune diseases (Fig. 7a, b).

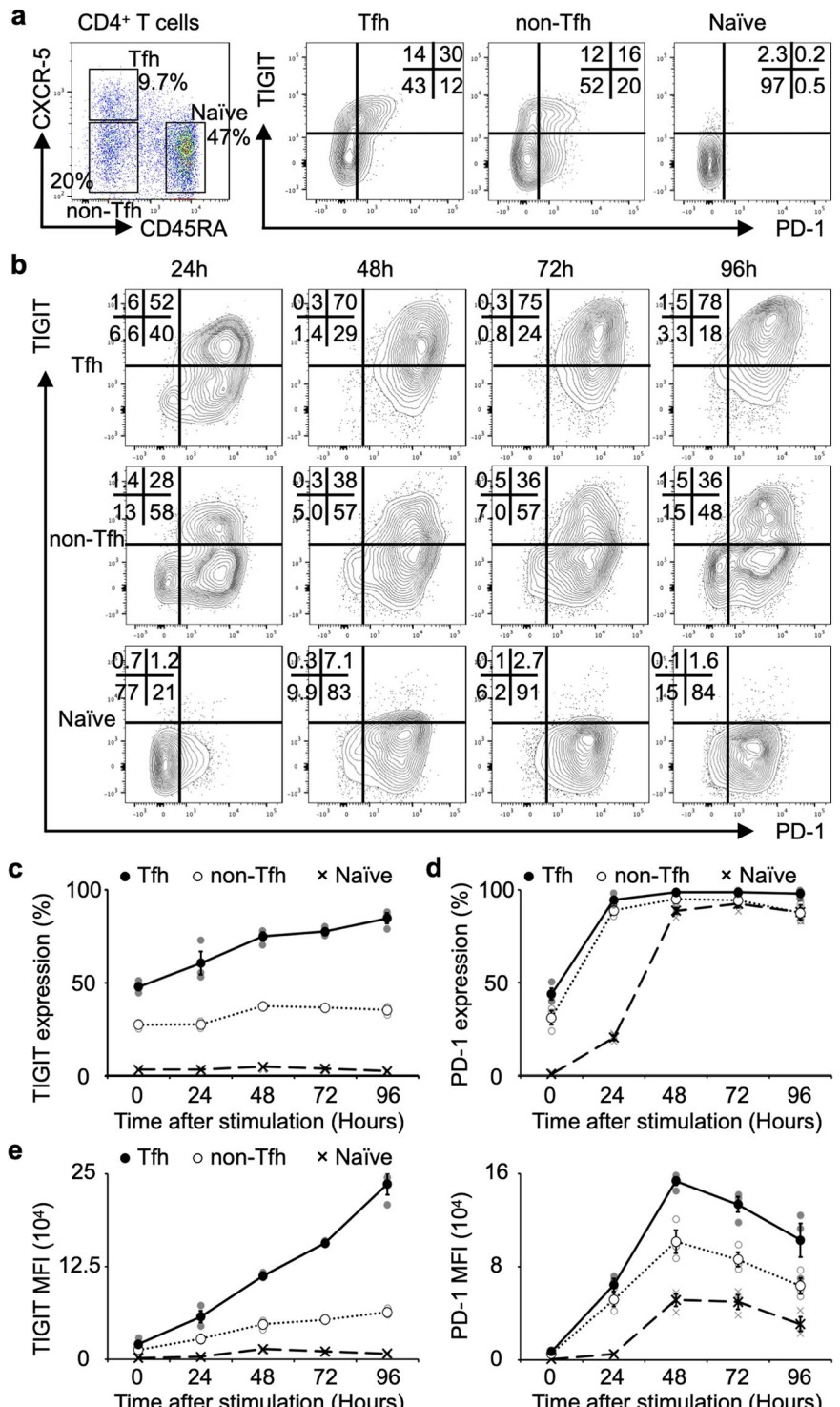

**Fig. 4 Activation induced TIGIT expression is specific in Tfh cells. a** Representative flow cytometry plots divided into CD45RA and CXCR5 staining of CD4+ T cells from HCs, and TIGIT and PD-1 expression in each divided subset is shown. **b–e** Each divided subset sorted from healthy donors ($n = 3$) and stimulated with anti-CD3/anti-CD28 beads. Samples were collected at the indicated time points, and TIGIT (**c**) and PD-1 (**d**) expression, and each mean fluorescence intensity (MFI) (**e**) (mean ± SEM) were assessed using flow cytometry.

We generated hu-*TIGIT* KI mice and used them to confirm the function of our mAb. In the past, an anti-CD28 superagonist mAb (TGN1412), which acts agonistically like our antibody, caused a life-threatening disaster[32]. One of the problems was that the mAb or its receptors used in the preclinical studies in animals were different from those used in the clinical studies. In addition, another problem was that the expression of CD28 was different

between humans and animals[33], and human lymphocytes are more reactive than those in chimpanzees to various stimuli[34]. This may suggest that it is insufficient to confirm the effect on animals as preclinical studies for the clinical application of novel agents. To avoid the TGN1412 disaster and accurately confirm the safety and function of our mAb, we created hu-*TIGIT* KI mice to confirm the function of the same mAb in vivo and in human

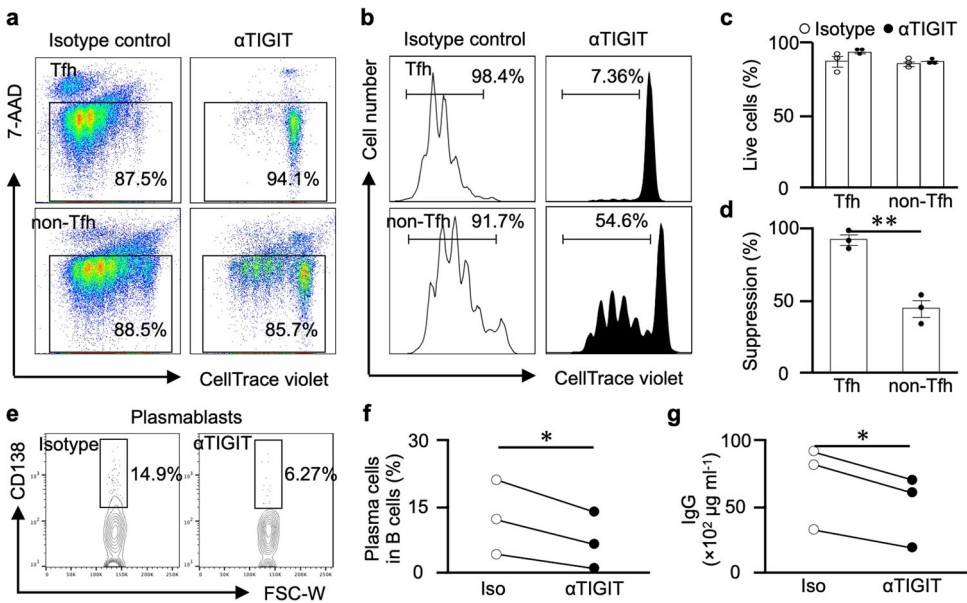

**Fig. 5 Anti-human-TIGIT agonistic mAb (αTIGIT) inhibits B cell activation via Tfh suppression.** Sorted T cells were labelled with CellTrace violet (CTV) and activated with anti-CD3 and anti-CD28 Abs with plate-bound αTIGIT (or the appropriate isotype control) for 4 days. CTV dilution was analysed by flow cytometry after gating on viable cells. Representative flow cytometry plots of viable cells (**a**) and CTV dilution (**b**) are shown. **c** The frequency of viable cells with αTIGIT compared to isotype control is shown. **d** The suppression rate (mean ± SEM) and the percentage of cell proliferation with αTIGIT compared to the isotype control are shown. **e**–**g** Tfh and CD27[+] memory B cells sorted from healthy donors ($n = 3$) were cocultured with αTIGIT (or the appropriate isotype control) and SEB for 7 days. **e** CD138 expression in plasma blast cells stimulated with αTIGIT (or the appropriate isotype control) in a representative donor. **f** Plasma cell differentiation ($n = 3$) and IgG production (**g**) ($n = 3$) are presented. $P$ values were determined by unpaired (**c**, **d**) or paired (**f**, **g**) two-tailed Student's $t$ test. $*P < 0.05$, $**P < 0.01$, $***P < 0.001$.

cells. As a preclinical study, our approach is better for assessing the safety and accurate function of agents in clinical applications.

Our mAb directly regulated T cells by stimulation with the inhibitory molecule TIGIT. From a similar perspective to our mAb, agonistic mAb against human PD-1, which is a coinhibitory molecule like TIGIT[24], is already used as a therapeutic target, and trials for autoimmune diseases have been conducted[35]. In addition, an anti-hu-PD-1 agonistic mAb has been reported to ameliorate airway hyperreactivity[36]. These results indicated that these mAbs suppressed T cell activation. On the other hand, PD-L1, a ligand for PD-1, has been reported to stabilize the expression of Foxp3 in induced Treg cells in vitro[31], but there are no reports that anti-hu-PD-1 agonistic mAbs enhance Treg cell function, similar to our mAb. Furthermore, PD-1 is expressed in a wide range of cells, and B cells, DCs, and monocytes are known to express only PD-1, not TIGIT[11,37]. For agonistic mAbs, the levels of receptor expression are important for safety and efficacy, and we analysed the expression of TIGIT and PD-1 in detail. Importantly, our study showed that TIGIT-expressing but not PD-1-expressing cells correlated with disease activity in autoimmune diseases. In addition, upon T cell stimulation, TIGIT-expressing cells were selectively upregulated among Tfh cells, while PD-1-expressing cells were non-specifically upregulated among CD4[+] T cells. From the above, we expect that our anti-hu-TIGIT agonistic mAb is safer and more effective than the anti-PD-1 agonistic mAb. However, these coinhibitory molecules are also expressed in cancer cells, and signalling inhibits the immune response, so blockades are already used in clinical cancer therapy[38,39]. It is unknown how our mAb affects malignancies, and further studies are required.

Our comprehensive analysis of autoimmune diseases proved that some subsets of TIGIT[+] CD4[+] T cells, especially TIGIT[+] Tfh and Tph cells, were associated with disease activity. In addition, our mAb had a stronger inhibitory effect on CD4[+]

T cells, especially Tfh and Tph cells, on which the expression of TIGIT was upregulated by stimulation. Our mAb acts on cells with elevated TIGIT expression, and this mechanism shows that the mAb can directly suppress the activation of autoreactive T cells in autoimmune diseases, even if the self-antigen is unknown. Furthermore, among activated CD4[+] T cells, our mAb strongly suppressed Tfh and Tph cells, which have already been reported to be involved in the production of autoantibodies[4,5] and are known to be associated with various autoimmune diseases[6–9,40–42]. In fact, in the IMQ study, our mAb strongly suppressed the production of autoantibodies, suggesting that it strongly suppresses the production of autoantibodies in autoimmune diseases. Agents that selectively inhibit Tfh and Tph cells have not been reported, and our mAb is expected to be a T cell targeted therapy and may be applicable to a wide range of autoimmune diseases by suppressing autoantibody production.

Our mAb not only suppressed non-Treg cells but also enhanced the suppressive function of Treg cells, especially FrI cells known as naive Treg cells[30]. Many reports have shown that Treg cell function is suppressed in autoimmune diseases[43–45], and enhancing Treg cell function is one of the challenges in the treatment of autoimmune diseases[46]. Abatacept (CTLA-4-Ig), which acts on T cells and is used as a costimulatory blockade, suppresses activated T cells, and its effect on Treg cell function is controversial[47,48]. Treg cell plasticity has been discussed[29], and Treg cells, especially naive Treg cells, are known to lose Foxp3 under inflammatory conditions[49]. It has been reported that Treg cells that lose Foxp3 expression convert into effector cells, affecting the pathogenesis of autoimmune diseases[50]. In a previous study using CD226 knockout mice, TIGIT stimulation of Treg cells increased their suppressive activity;[51] our results were consistent with these previous results. In addition, our results showed that our mAb retains Foxp3[high] Treg cell proportions, which are downregulated under inflammatory conditions, in vivo.

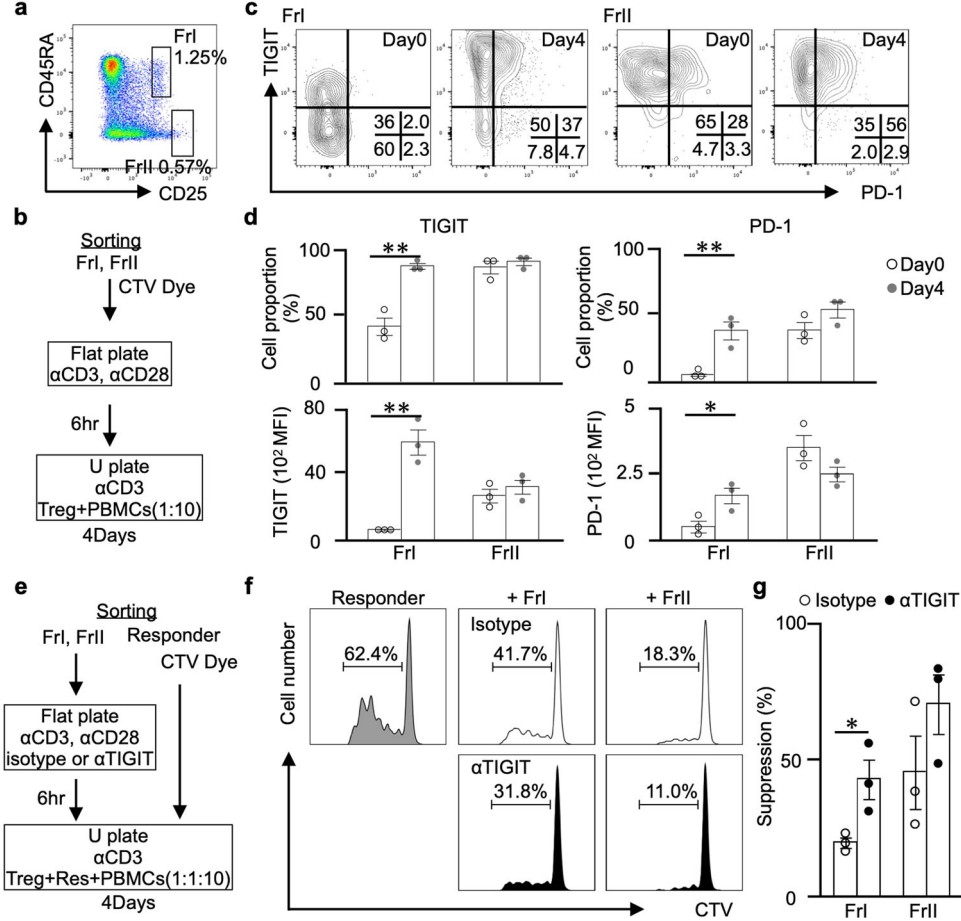

**Fig. 6 Anti-human-TIGIT agonistic mAb (αTIGIT) enhances suppressive activity of naive Treg cells. a** Representative flow cytometry plots for Treg subtypes divided into CD25 and CD45RA staining of CD4+ T cells and each indicated fraction from HCs are shown. **b** Each Treg subset stimulation protocol is shown. **c** TIGIT and PD-1 expression in each Treg subtype before and after stimulation in a representative donor. **d** The proportion and MFI of TIGIT and PD-1 expression in each Treg subtype in unstimulated and stimulated HCs ($n = 3$) is shown. **e** The coculture protocol for stimulated Treg subsets and responder T cells is shown. **f** CellTrace violet (CTV) dilution, which was analysed by flow cytometry after gating on CTV-labelled responder T cells, and the suppression rate (**g**) was calculated as follows: (1−(proliferation of cocultured responder T cell)/(proliferation of cultured responder T cell alone)) × 100) (mean ± SEM) are shown. The P values were determined using an unpaired two-tailed Student's t-test. *P < 0.05, **P < 0.01, ***P < 0.001.

We also demonstrated for the first time that our mAb enhances the function of human Treg cells, especially FrI cells, in vitro. These results indicate that our mAb is useful for directly acting on Treg cells, whose function is suppressed in autoimmune diseases, to maintain or enhance their function.

Our use of KI mice enabled us to demonstrate the effect of each mAb type both in vitro and in vivo, but this aspect of the study was associated with a certain limitation: The hu-TIGIT expression level of Tfh cells in our KI mice was approximately one-eighth that of the mouse-TIGIT expression level in WT mice and one-twentieth of the hu-TIGIT expression level in human cells (Figs. 1a and Fig. 3c). Nevertheless, we were able to confirm that the effects of our mAb on T cells were the same in vitro and in vivo and to demonstrate the efficacy of this mAb in hu-TIGIT KI mice. We consider this is because hu-TIGIT expression in KI mice increased to the same level as mo-TIGIT in WT mice under inflammatory conditions (Supplementary Fig. 5a, b). Thus, we believe it is enough to assess function of our mAb in this KI mice. Furthermore, this low expression in hu-TIGIT KI mice suggests that our mAb may be even more effective in humans, as humans express higher levels of TIGIT.

There is another limitation of our study. It has been reported that mouse CD155 can bind to hu-TIGIT and convey the inhibitory signal[52]. We do not directly assess this binding in vivo.

However, even if the mouse CD155 interacts hu-TIGIT, we believe that our experiments were able to prove the effect of the mAb itself by using an untreated group as a control.

In summary, we demonstrated the function and efficacy of our mAb both in vitro and in vivo by utilizing hu-TIGIT KI mice and human T cells. We have shown that our mAb selectively and strongly inhibits the activation of CD4+ T cells, especially Tfh and Tph cells, and is simultaneously capable of enhancing FrI cell function. These features reveal that this mAb strongly improves the imbalance of T cells in autoimmune diseases by acting on two different types of cells with different effects and is likely to be particularly useful in treating autoimmune diseases.

## Methods

**Patients**. We prospectively enrolled a total of 10 untreated and 5 active patients with RA, 10 patients with active SLE, 20 patients with untreated SjS and 15 HCs who visited Keio University Hospital from 2018 to 2020. Each patient fulfilled the 2010 American College of Rheumatology/EULAR classification criteria for RA[53], the 1997 American College of Rheumatology criteria for SLE[54], or the 2016 American College of Rheumatology/EULAR criteria for SjS[55], We defined active RA as having a disease activity of DAS28-CRP >2.7 and active SLE as requiring additional treatment by the physician. Various parameters for each patient were obtained from electronic medical records and are shown in Supplementary Tables 1 and 2. The study was approved by the Institutional Review Board of Keio University School of Medicine (No. 20110258) and conducted in compliance with the tenets of the Declaration of Helsinki. Written informed consent was obtained

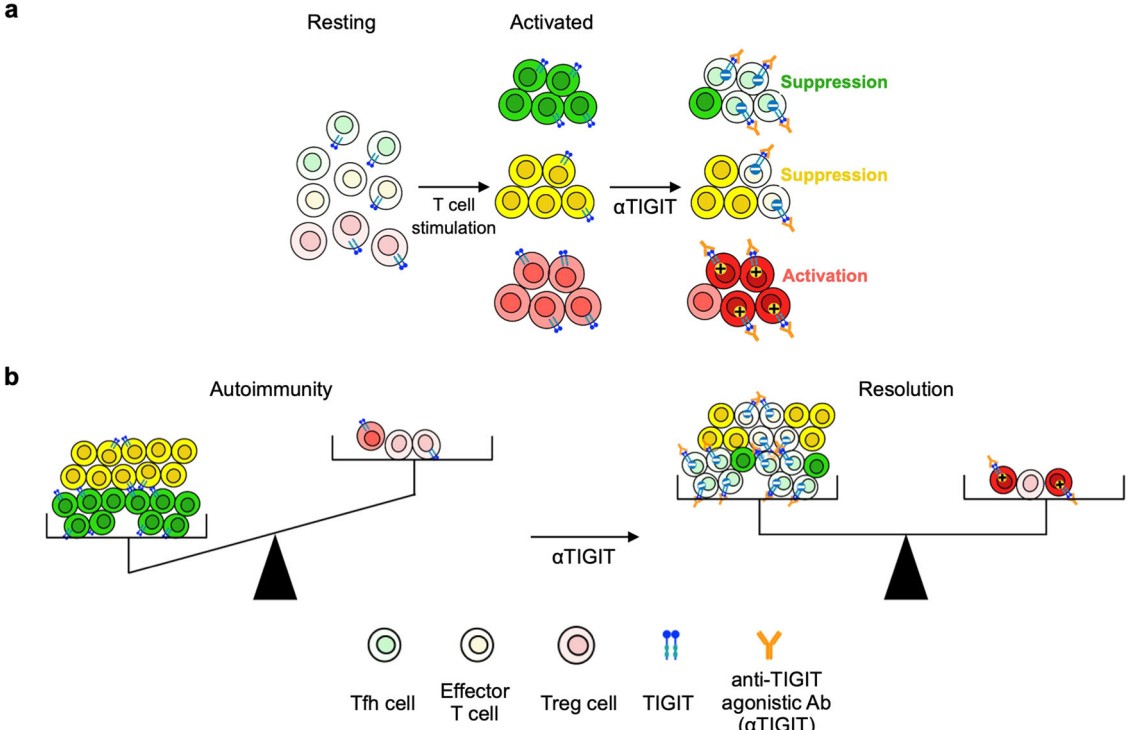

**Fig. 7 Anti-human-TIGIT agonistic mAb (αTIGIT) regulates T cell imbalance in autoimmune states. a** TIGIT expression is upregulated by T cell stimulation, especially in Tfh and Treg cells. αTIGIT selectively and strongly suppressed Tfh cells and promoted the suppressive function of Treg cells. **b** αTIGIT can recover imbalance towards activated T cells by conflicting effects on CD4$^+$ T cells.

from all participating individuals. PB samples for immunophenotyping were collected from patients seen at Keio University Hospital.

**Mice**. All the mice were used at 6–8 and 10–15 weeks of age. Animal experiments were performed in strict accordance with the recommendations in the Guidelines for Proper Conduct of Animal Experiments of the Science Council of Japan. All the experiments were approved by the Institutional Animal Research Committee and Ethics Committee of Keio University (No. 17071).

**hu-*TIGIT* KI mice**. hu-*TIGIT* KI mice on the C57BL/6JJcl background were generated at Takeda Pharmaceutical Company, Limited (Kanagawa, Japan). Hu-*TIGIT* KI mice were generated by homologous recombination in mouse embryonic stem (ES) cells. The 0.8 kb cDNA fragment corresponding to human *TIGIT* was amplified by PCR using the following primers: 5′-ATGAGATGGTGCCTGCTG CTG-3′ and 5′-TCAGCCTGTCTCGGTGAAGAAGC-3′. The PCR product was assembled upstream of the SV40 polyA signal (SpA) with an FRT-neomycin resistance gene cassette. The assembled sequence was flanked with the homology arms for replacement of the mouse *Tigit* gene to create a targeting vector. The linearized targeting vector was electroporated into C57BL/6J ES cells. The targeted ES cell clones were injected into ICR blastocysts after neomycin selection. The resulting chimeric mice were genotyped by PCR with the following primer set: 5′-GGTCATTCATTTGGGTGCCTGTAAA-3′ and 5′-GGTAGGTGTGGTAGATGC AGAAGTA-3′. WT C57BL/6JJcl mice (6–8 weeks) were purchased from Clea Japan, Inc. (Tokyo, Japan) and bred in-house. The same generation of hu-*TIGIT* KI mice and C57BL/6JJcl mice was used in each experiment.

**Generation of anti-hu-TIGIT agonistic mAbs**. The mAbs were developed at Takeda Pharmaceutical Company, Limited[56]. In all, 6- to 8-week-old male CD2F1 mice were immunized by subcutaneous administration of recombinant human TIGIT-His-mFc protein (10 μg mouse$^{-1}$, in TiterMax), followed by booster immunization in the popliteal fossae. The collected lymph node cells were fused with P3-X63.Ag8. U1 (P3U1) cells using a Hybrimune Electrofusion System (BTX, MA, USA), and the hybridomas were selected in ClonaCell$^{TM}$-HY Medium E (Stemcell Technologies, Vancouver, Canada). Culture supernatants were screened for specificity by ELISA and flow cytometry as follows.

**Anti-hu-TIGIT agonistic mAb biological activity assays**. For the agonist assay, Jurkat T cells engineered to express human TIGIT with the cells containing a plasmid with luciferase under the control of human TIGIT were cultured with anti-hu-TIGIT agonistic mAbs (100 ng ml$^{-1}$ to 10 μg ml$^{-1}$), PVR (R&D Systems, MN, USA) (333 ng ml$^{-1}$ to 33 μg ml$^{-1}$), an isotype control (100 ng ml$^{-1}$ to 10 μg ml$^{-1}$),

or a blank in RPMI 1640 medium supplemented with 10% FBS (Corning, NY, USA), 100 U ml$^{-1}$ penicillin (Fujifilm Wako Pure Chemical Corporation, Tokyo, Japan), and 100 U ml$^{-1}$ streptomycin (Fujifilm Wako Pure Chemical Corporation) and maintained at 37 °C in a 5% $CO_2$ humidified atmosphere for 24 h (day 1). Then, the cells were collected and cultured in 96-well flat-bottom plates coated with anti-CD3 (3 μg ml$^{-1}$) and containing soluble anti-CD28 (2 μg ml$^{-1}$) for 24 h (day 2). Finally, the luminescence values of the cultured cells were measured with a Nano-Glo Luciferase Assay System (Promega, WI, USA), and the agonist activity (%) was calculated as (1 − ((values of sample) − (values of 100% control (PVR 3.3 μg ml$^{-1}$)))/((values of 0% control (blank)) − (values of 100% control (PVR 3.3 μg ml$^{-1}$))) × 100).

For the antagonist assay, Jurkat T cells engineered to express human TIGIT with a luciferase reporter were cultured with the anti-hu-TIGIT agonistic mAbs (100 ng ml$^{-1}$ to 10 μg ml$^{-1}$), PVR (R&D Systems) (333 ng ml$^{-1}$ to 33 μg ml$^{-1}$), an isotype control (100 ng ml$^{-1}$ to 10 μg ml$^{-1}$), or a blank, and PVR at a 3.3 μg ml$^{-1}$ final concentration was added without a blank and maintained at 37 °C in a 5% $CO_2$ humidified atmosphere for 24 h (day 1). Luminescence values were measured on day 2 via the same procedure described for the agonist assay; the antagonist activity was calculated as (((values of sample) − (values of 100% control (blank)))/ ((values of 0% control (3.3 μg ml$^{-1}$ PVR)) − (values of 100% control (blank))) × 100).

For the cytotoxicity/growth inhibition assay, Jurkat cells were prepared and treated as described for days 1 and 2 in the agonist assay. The luminescence values of the cultured cells were measured with the CellTiter-Glo Luminescent Cell Viability Assay System (Promega), and cytotoxic activity was calculated as (1 − ((values of sample) − (values of 0% control (blank)))/(values of 100% control (3.3 μg ml$^{-1}$ PVR)) − (values of 0% control (blank))) × 100).

**Cross-recognition of M1-8**. We cultured hu-TIGIT JI tet CHO, macaque-TIGIT T-REx-CHO, or mo-TIGIT T-REx-CHO cells (in-house) with or without doxycycline (TAKARA BIO, Shiga, Japan) for 2 days. Those cells were incubated with M1-8 (1 μg ml$^{-1}$) for 30 min at 4 °C and washed three times with staining buffer (PBS mixed with 1% FBS (Thermo Fisher Scientific, MA, USA)). Then, we corrected the samples and stained them with anti-mouse IgG (AlexaFluoro647) (Jackson ImmunoResearch, PA, USA) antibodies for 30 min at 4 °C. The samples were washed twice with staining buffer and analysed with a Cytomics FC500MPL (Beckman Coulter, CA, USA).

**Induction of EAE**. 12-week-old male hu-*TIGIT* KI mice were subcutaneously (s.c.) injected with 200 μg of MOG35–55 peptide (Synpeptide, Shanghai, China) and 400 μg of *Mycobacterium tuberculosis* in incomplete Freund's adjuvant (Difco Laboratories, NJ, USA). Pertussis toxin (400 ng per mouse; List Biologicals, CA,

USA) was intraperitoneally injected the same day and on day 2 post immunization. We also intraperitoneally administered the anti-hu-TIGIT agonistic mAb (100 µg per mouse in 100 µl PBS) or a mouse IgG1κ isotype control (clone MG1-45, BioLegend, 100 µg per mouse, in 100 µl PBS) to each mouse on days 0, 2, 4, 10, and 17. The animals were observed daily for clinical symptoms and scored by a blinded investigator as follows: 0, no clinical disease; 1, tail weakness; 2, tail paralysis; 3, hindlimb weakness; 4, forelimb weakness; 5, forelimb paralysis; and 6, moribundity or death.

**Induction of the IMQ-induced lupus mouse model.** Female hu-*TIGIT* KI mice (10–15 weeks old) were treated topically with 5% imiquimod cream (Mochida Pharmaceutical, Tokyo, Japan) on their right ear at 50 mg kg$^{-1}$ three times per week during the study. We administered anti-TIGIT agonistic mAb (300 µg per mouse in 100 µl PBS) or a mouse IgG1κ isotype control (300 µg per mouse in 100 µl PBS) intraperitoneally twice per week during the study. The mice were sacrificed on day 42 after disease induction for flow cytometric analysis of spleens and lymph nodes.

**Cell preparation.** Mouse spleens were removed and weighed. Each spleen was minced, and erythrocytes were lysed with HLB solution (IBL, Gunma, Japan). After washing twice with complete RPMI medium (RPMI 1640 medium supplemented with 2 mM L-glutamine, 1% nonessential amino acids, 55 µM 2-mercaptoethanol (Thermo Fisher Scientific), 1 mM sodium pyruvate, 100 U ml$^{-1}$ penicillin, 100 U ml$^{-1}$ streptomycin, and 10% FBS (Thermo Fisher Scientific)) and filtering through a 40-µm cell strainer, the total splenocyte count was recorded with Trypan Blue (Nacalai Tesque, Kyoto, Japan). The processed lymph nodes were also passed through a 40-µm cell strainer. Then, these cells were immediately analysed with a FACSAria III flow cytometer (BD Biosciences, NJ, USA).

**Quantification of anti-dsDNA antibodies.** Blood samples were collected from the lupus mice into EDTA-containing tubes. After centrifugation, we collected the plasma, and the levels of anti-dsDNA antibodies were measured using the Levis anti-dsDNA-mouse ELISA kit (Shibayagi, Gunma, Japan) according to the manufacturer's instructions.

**Immunophenotyping.** PB samples were collected from all the enrolled patients and HCs at Keio University Hospital. We collected 100 µl of blood into a heparin blood collection tube (Terumo, Tokyo, Japan) and incubated the blood with antibodies (indicated below and in Supplementary Table 3) for 15 min. Then, we added 1.5 ml of FACS Lysing Solution (BD Biosciences) to fix the cells and lyse red blood cells. After 10 min, the samples were washed twice with staining buffer (PBS mixed with 0.5% BSA and 2 mM EDTA) and analysed with a FACSAria III flow cytometer. The data were analysed with FlowJo software version 10.4.2 (FlowJo, OR, USA). We defined the developmental stages as follows: Tn, CCR7$^+$ CD45RA$^+$; Tcm, CCR7$^+$ CD45RA$^-$; Tem, CCR7$^-$ CD45RA$^-$; Temra, CCR7$^-$ CD45RA$^+$; Tfh, CD45RA$^-$ CXCR5$^+$; activated Tfh, CCR7$^{low}$ CD45RA$^-$ CXCR5$^+$ PD-1$^{high}$; Tph, CCXR5$^-$ PD-1$^{high}$; Treg, CD25$^+$ CD127$^-$; FrI, CD25$^+$ CD45RA$^+$; and FrII, CD25$^{high}$ CD45RA$^-$. The gating strategies are shown in Supplementary Figs. 8 and 9.

For splenocyte, lymphocyte, and thymocyte surface staining, the cells were incubated with anti-CD16/CD32 (BD Biosciences) for 5 min at 4 °C and surface stained with antibody panels (see below and Supplementary Table 4) for 20 min with staining buffer. For intracellular staining, the cells were first incubated for 4 h in complete RPMI medium containing 1 µg ml$^{-1}$ ionomycin, 1 µg ml$^{-1}$ PMA, and GolgiPlug (BD Biosciences) and then surface stained as described above. Then, the Foxp3/Transcription Factor Staining Buffer Set (Thermo Fisher Scientific) was used for intracellular staining according to the manufacturer's instructions. Each stained sample was washed twice with staining buffer and analysed with a FACSAria III flow cytometer. The data were analysed with FlowJo software version 10.4.2. We defined developmental stages as follows: Tn, CD44$^-$ CD62L$^+$; Tcm, CD44$^+$ CD62L$^+$; Tem, CD44$^+$ CD62L$^-$; Tfh, CXCR5$^+$ PD-1$^+$; Tph, CXCR5$^-$ PD-1$^{high}$; Treg, Foxp3$^+$ or Foxp3$^{high}$; GC B, B220$^+$ CD95$^+$ GL7$^+$; and plasma, CD19$^{low}$ CD138$^+$. The gating strategies are shown in Supplementary Figs. 10 and 11.

**Cell sorting.** PBMCs were isolated from heparinized blood from the HCs via density gradient centrifugation (Lymphoprep Axis-Shield, Oslo, Norway). CD4$^+$ or CD19$^+$ T cells were isolated by negative or positive selection (CD4$^+$ T Cell or CD19$^+$ Cell Isolation Kit, Miltenyi Biotec, Bergisch-Gladbach, Germany) as needed. The cells were stained with anti-human antibodies against the markers CD4 (BV510 or FITC), CD25 (PE), CD27 (BV510), CD45RA (BV421), CD127 (APC), CXCR5 (PerCP-Cy5.5), and TIGIT (PE-Cy7) as needed for 20 min with staining buffer. Each stained sample was washed twice with staining buffer and sorted using a FACSAria III flow cytometer. The data were analysed with FlowJo software version 10.4.2. We defined the developmental stages as follows: Tn, CD45RA$^+$ CXCR5$^-$; Tfh, CD45RA$^-$ CXCR5$^+$; non-Tfh, CD45RA$^-$ CXCR5$^-$; responder,

CD25$^-$; FrI, CD25$^+$ CD45RA$^+$; FrII, CD25$^{high}$ CD45RA$^-$; and CD27$^+$, memory B. The gating strategies are shown in each section.

**TIGIT expression of Tfh cells (human).** Sorted CD45RA$^-$ CXCR5$^+$ Tfh, CD45RA$^-$ CXCR5$^-$ non-Tfh, and CD45RA$^+$ CXCR5$^-$ naive cells (1 × 10$^4$ cells each) were cultured with the same number of Dynabeads conjugated to Human T-Activator CD3/CD28 (VERITAS, Tokyo, Japan) in complete RPMI medium and maintained at 37 °C in a 5% CO$_2$ humidified atmosphere. Every 24 h for 4 days, each cell was stained with anti-human TIGIT-PE-Cy7 and anti-human PD-1-APC-Cy7 for 20 min with staining buffer. Each stained sample was washed twice with staining buffer, and live cells gated on 7-aminoactinomycin D (BioLegend, CA, USA) were analysed on a FACSAria III flow cytometer. The data were analysed with FlowJo software version 10.4.2.

**In vitro Tfh cell suppression assay (human).** The sorted Tfh, non-Tfh, and naive cells were labelled with 4 µM CTV, and 1 × 10$^4$ cells per well were cultured in duplicate in 96-well flat-bottom plates coated with anti-human CD3 (2 µg ml$^{-1}$), anti-human CD28 (1 µg ml$^{-1}$) and anti-hu-TIGIT agonistic mAb (10 µg ml$^{-1}$) or a mouse IgG1κ isotype control (10 µg ml$^{-1}$) in complete RPMI medium for 4 days. Then, we analysed live cells gated on 7-aminoactinomycin D and CTV dilution by each cell type using a FACSVerse flow cytometer (BD Biosciences) and calculated the percent suppression as ((proliferation rete of mAb)/(proliferation rete of isotype) × 100). The data were analysed with FlowJo software version 10.4.2.

**In vitro Tfh-B cell coculture assay (human).** The sorted 1 × 10$^4$ Tfh cells were cultured in 96-well U-bottom plates coated with anti-hu-TIGIT agonistic mAb (10 µg ml$^{-1}$) or a mouse IgG1κ isotype control (10 µg ml$^{-1}$) for 30 min and then cocultured with 1 × 10$^4$ sorted memory B cells in complete RPMI medium containing 0.2 µg ml$^{-1}$ SEB for 7 days. Then, we corrected the cells and stained them with anti-human antibodies against the markers CD27 (BV510), CD38 (FITC), and CD138 (APC) for 20 min with staining buffer, and each stained sample was washed twice with staining buffer. Then, we analysed the percentage of plasma cells (defined as CD27$^+$ CD38$^+$ CD138$^+$) using a FACSVerse flow cytometer and measured total human IgG in the supernatant using an IgG ELISA kit (Bethyl, TX, USA) according to the manufacturer's instructions. The data were analysed with FlowJo software version 10.4.2.

**In vitro Treg cell activation assay (human).** The sorted Treg cells were labelled with CTV and cultured separately in a 96-well flat-bottom plate coated with anti-human CD3 (2 µg ml$^{-1}$) and anti-human CD28 (1 µg ml$^{-1}$) in complete RPMI medium for 6 h. Then, each Treg cell was cocultured with 1 × 10$^5$ irradiated PBMCs (30 Gy) of the same HCs in duplicate in a 96-well U-bottom plate coated with anti-human CD3 (2 µg ml$^{-1}$) in complete RPMI medium for 4 days. However, for FrII, we collected so small number of samples that we could not perform in duplicate. The accuracy was evaluated by repeated. At the end of the culture period, the cells were stained with anti-human TIGIT (PE-Cy7) and anti-human PD-1 (APC-Cy7) for 20 min with staining buffer, and each stained sample was washed twice with staining buffer. Then, CTV-positive cells were analysed for marker expression using a FACSVerse flow cytometer. The data were analysed with FlowJo software version 10.4.2.

**In vitro Treg-responder T cell coculture assay (human).** The sorted Treg cells (FrI, FrII; 1 × 10$^4$) were cultured in 96-well flat-bottom plates coated with anti-human CD3 (2 µg ml$^{-1}$), anti-CD28 (1 µg ml$^{-1}$) and anti-TIGIT agonistic mAb (10 µg ml$^{-1}$) or a mouse IgG1 isotype control (10 µg ml$^{-1}$) in complete RPMI medium for 6 h. Then, a total of 1 × 10$^4$ Treg cells were cocultured with 1 × 10$^4$ CTV-labelled CD25$^-$ responder T cells and 1 × 10$^5$ irradiated PBMCs (30 Gy) from the same HCs in duplicate in 96-well flat-bottom plates coated with anti-human CD3 (2 µg ml$^{-1}$) in complete RPMI medium for 4 days. However, for FrII, we collected so small number of samples that we could not perform in duplicate. The accuracy was evaluated by repeated. After the incubation period, CTV-positive responder cells were analysed for CTV dilution using a FACSVerse flow cytometer, and the suppression rate was calculated as (=(1 − (proliferation rate of cocultured responder T cells)/(proliferation rate of cultured responder T cells alone)) × 100). The data were analysed with FlowJo software version 10.4.2.

**Antibodies and reagents.** The antibodies used are shown in Supplementary Tables 3 and 4.

**Statistics and reproducibility.** There were no statistical tests to determine sample size. The sample size was determined from the results of a preliminary study. The number used for each experiment is shown in figure legend. Multiple independent studies confirmed consistent results. In vitro, experiments were performed in duplicate and repeated at least twice independently; in vivo, experiments were also performed twice independently. The data are shown as the mean ± SEM and were analysed using JMP 15 (SAS Institute, NC, USA). Statistical analyses were

performed by using the Wilcoxon rank sum test or paired or unpaired two-tailed Student's $t$ test for two-group comparisons and Spearman's rank for correlation analysis, all as described in the figure legends. $P$ values less than 0.05 were considered significant.

## Data availability

All data related to the study have been included in the article or uploaded as supplementary information. Those source data can be found in Supplementary data. Other original data of this study are available on reasonable request to the corresponding authors (TT).

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

## Acknowledgements

We thank Ms. Yukari Kaneda, Ms. Yumi Ikeda, Ms. Kumiko Tanaka, and Ms. Harumi Kondo for assisting with the experiments. This work was supported by JSPS KAKENHI 21H05044, JSPS KAKENHI (C) JP20K08806, JSPS KAKENHI JP20K22789, AMED-CREST JP21gm1110009 and Moonshot JP21zf0127003h0001.

## Author contributions

Conceptualisation: M.K., K.S., M.I., H.Y., K.K., Yo.K., Yu.K., R.M., A.Y., and T.T. Funding acquisition: M.O., K.S., A.Y., and T.T. Investigation: M.K., K.S., M.T., M.O., M.I., H.Y., K.K., T.K., Yo.K., T.Y., R.A., K.H., Y.O., and K.Y. Methodology: M.K., K.S., M.T., M.O., M.I., H.Y., K.K., T.Y., Yo.K., T.K., R.A., K.H., Y.O., K.Y., Yu.K., R.M., A.Y., and T.T. Project administration: K.S., K.K., K.Y., A.Y., and T.T. Resources: M.K., K.S., M.T., M.O., M.I., H.Y., K.K., Y.O., K.Y., Yu.K., R.M., A.Y., and T.T. Supervision: K.S., A.Y., and T.T. Writing-original draft preparation: M.K. and K.S. Writing-review and editing: M.K., K.S., M.T., M.O., M.I., H.Y., K.K., Yo.K., Y.O., K.Y., Yu.K., R.M., A.Y., and T.T.

## Competing interests

K.K., T.K., Yo.K., T.Y., R.A., and K.H. are employees of Takeda Pharmaceutical Company Limited. This study was performed as part of a research collaboration between Keio University and Takeda Pharmaceutical Company Limited. Authors M.K., K.S., M.T., K.K., T.K., Yo.K., T.Y., R.A., A.Y. and T.T. hold the patent (Patent # WO2022/075474) for anti-human-TIGIT agonistic antibody. The remaining authors declare no competing interests.
