## [Peer Review File · Communications Biology]

Reviewers' comments:

Reviewer #1 (Remarks to the Author):

Co-inhibitory molecules have an enormous potential as therapeutic targets in tumor therapy but also in autoimmune diseases, where they may be able to restore immune homeostasis by dampening autoimmune responses and enhancing Treg function. In this study, Kojima et al generate agonistic antibodies against the co-inhibitory receptor TIGIT and a new mouse model that allows for the in vivo assessment of therapeutic that target human TIGIT to be assessed. In a first step, the authors confirm an elevated expression of TIGIT in patients with RA and SLE. They then go on to raise agonistic anti-human TIGIT antibodies which they then functionally test in vivo in the human TIGIT KI mouse model they newly develop. The agonistic anti-TIGIT Ab is able to suppress disease severity in the EAE model and disease parameters in the imiquimod-induced lupus model. They further show that their Ab suppresses Tfh function but enhances Treg-mediated suppression in vitro. Overall, this is an interesting study and the developed mouse model will likely be very helpful to test new tools developed to target human TIGIT. Nevertheless, several aspects of the study should be improved:

Major point:

1. While the mouse model allows for the expression of human TIGIT, the expression levels are strongly reduced. The reasons for and implications of this should be investigated and discussed. Rather than focusing on Tfh, which are rare in naïve mice, (Fig. 2c) the authors should also analyze the expression of ms vs hu TIGIT in Tregs. This should give a clearer impression of the expression levels. Furthermore, it would be important to provide statistics on the expression levels (Suppl. Fig. 3) and perform a proper phenotyping of this newly developed mouse model (T cell development and numbers, lymphocyte populations, etc). How well does mouse CD155 bind to human TIGIT? Does reduced binding/stimulation perhaps lead to reduced TIGIT activity resembling a KO? Does this manifest in changes in disease susceptibility (e.g. the EAE scores seem relatively high)? Also, why did the authors use cDNA (rather than the full human gene) in their construct? Why did they not include the human promoter in the KI, which would have been more representative of the TIGIT expression pattern in human? This should be discussed.

Additional points:

2. In Fig. 4 the authors investigate TIGIT expression upon stimulation. While the change in frequency of TIGIT expressing cells are limited, it seems like there are large differences in the MFI. It would therefore be important to also analyze the changes in MFI.
3. It is difficult to understand how the assay to test the biological activity of the Abs (Suppl. Fig. 2) was performed. It would be helpful if the authors included a schematic in the supplementary information.

Minor points:

4. It is difficult to understand how the assay to test the biological activity of the Abs (Suppl. Fig. 2) was performed. It would be helpful if the authors included a schematic in the supplementary information.
5. There is a misleading typo/error on page 6 in the first paragraph, where it read '... our mAb significantly suppressed splenomegaly and REDUCED the number of splenocytes...'.
6. Why did the authors use such a high dose of the Ab in the IMQ model for SLE?
7. The gating strategy for the Foxp3hi Tregs should be included in the supplementary figures.
8. The last sentence of the second paragraph on page 7 should read most pronounced rather than most selective as the effect is not exclusive.
9. On page 7 the authors state that there is a stronger inhibitory effect of the anti-TIGIT Ab on Tfh than on non-Tfh. While this is true, it would be important to mention that the non-Tfh cells are also strongly suppressed.

Reviewer #2 (Remarks to the Author):

Communications Biology manuscript TITLE: "Anti-human-TIGIT agonistic antibody ameliorate autoimmune diseases by restoring T cell imbalance" presented interesting findings. The strengths of the manuscript included generation of anti-human -TIGIT agonistic monoclonal antibodies and human TIGIT knock-in mice, and findings of the mAbs' advantages in inhibiting autoimmune T cell responses and enhancing immunosuppressive Treg function. However, the manuscript needs to be improved by fixing the following major concerns in presentation and analysis of the results.

There are major concerns:

1. The last part of manuscript title is misleading. The last part of the title can be revised into "...by inhibiting autoimmune Tfh and pTh cells and enhancing Tregs";
2. The authors did not present any significant findings from human TIGIT knock-in mice in the Abstract and the Results section;
3. The authors need to use summary sentences as the titles for all the figures;
4. All the figures need to be self-telling so that reviewers and readers understand the figures right away, rather than letting them looking around and trying to understand the figures.

Reviewers' comments:

Reviewer #1 (Remarks to the Author):

Co-inhibitory molecules have an enormous potential as therapeutic targets in tumor therapy but also in autoimmune diseases, where they may be able to restore immune homeostasis by dampening autoimmune responses and enhancing Treg function. In this study, Kojima et al generate agonistic antibodies against the co-inhibitory receptor TIGIT and a new mouse model that allows for the in vivo assessment of therapeutic that target human TIGIT to be assessed. In a first step, the authors confirm an elevated expression of TIGIT in patients with RA and SLE. They then go on to raise agonistic anti-human TIGIT antibodies which they then functionally test in vivo in the human TIGIT KI mouse model they newly develop. The agonistic anti-TIGIT Ab is able to suppress disease severity in the EAE model and disease parameters in the imiquimod-induced lupus model. They further show that their Ab suppresses Tfh function but enhances Treg-mediated suppression in vitro.

Overall, this is an interesting study and the developed mouse model will likely be very helpful to test new tools developed to target human TIGIT. Nevertheless, several aspects of the study should be improved:

Major point:

1. While the mouse model allows for the expression of human TIGIT, the expression levels are strongly reduced. The reasons for and implications of this should be investigated and discussed. Rather than focusing on Tfh, which are rare in naïve mice, (Fig. 2c) the authors should also analyze the expression of ms vs hu TIGIT in Tregs. This should give a clearer impression of the expression levels. Furthermore, it would be important to provide statistics on the expression levels (Suppl. Fig. 3) and perform a proper phenotyping of this newly developed mouse model (T cell development and numbers, lymphocyte populations, etc). How well does mouse CD155 bind to human TIGIT? Does reduced binding/stimulation perhaps lead to reduced TIGIT activity resembling a KO? Does this manifest in changes in disease susceptibility (e.g. the EAE scores seem relatively high)? Also, why did the authors use cDNA (rather than the full human gene) in their construct? Why did they not include the human promoter in the KI, which would have been more representative of the TIGIT expression pattern in human? This should be discussed.

Additional points:

2. In Fig. 4 the authors investigate TIGIT expression upon stimulation. While the change in frequency of TIGIT expressing cells are limited, it seems like there are large differences in the MFI. It would therefore be important to also analyze the changes in MFI.
3. It is difficult to understand how the assay to test the biological activity of the Abs (Suppl. Fig. 2) was performed. It would be helpful if the authors included a schematic in the supplementary information.

Minor points:

4. It is difficult to understand how the assay to test the biological activity of the Abs (Suppl. Fig. 2) was performed. It would be helpful if the authors included a schematic in the supplementary information.
5. There is a misleading typo/error on page 6 in the first paragraph, where it read '... our mAb significantly

suppressed splenomegaly and REDUCED the number of splenocytes...’.

6. Why did the authors use such a high dose of the Ab in the IMQ model for SLE?

7. The gating strategy for the Foxp3hi Tregs should be included in the supplementary figures.

8. The last sentence of the second paragraph on page 7 should read most pronounced rather than most selective as the effect is not exclusive.

9. On page 7 the authors state that there is a stronger inhibitory effect of the anti-TIGIT Ab on Tfh than on non-Tfh. While this is true, it would be important to mention that the non-Tfh cells are also strongly suppressed.

Reviewer #2 (Remarks to the Author):

Communications Biology manuscript TITLE: “Anti-human-TIGIT agonistic antibody ameliorate autoimmune diseases by restoring T cell imbalance” presented interesting findings. The strengths of the manuscript included generation of anti-human -TIGIT agonistic monoclonal antibodies and human TIGIT knock-in mice, and findings of the mABs’ advantages in inhibiting autoimmune T cell responses and enhancing immunosuppressive Treg function. However, the manuscript needs to be improved by fixing the following major concerns in presentation and analysis of the results.

There are major concerns:

1. The last part of manuscript title is misleading. The last part of the title can be revised into“...by inhibiting autoimmune Tfh and pTh cells and enhancing Tregs”;
2. The authors did not present any significant findings from human TIGIT knock-in mice in the Abstract and the Results section;
3. The authors need to use summary sentences as the titles for all the figures;
4. All the figures need to be self-telling so that reviewers and readers understand the figures right away, rather than letting them looking around and trying to understand the figures.

30th June, 2022

Dear Reviewers,

Thank the editor and reviewers for reviewing and improving our manuscript (COMMSBIO-22-0736) titled "Anti-human-TIGIT agonistic antibody ameliorates autoimmune diseases by restoring T cell imbalance" and their detailed and constructive feedback. The following are the point-by-point response to the reviewers' comments.

Reviewers' comments:

Reviewer #1 (Remarks to the Author):

Co- inhibitory molecules have an enormous potential as therapeutic targets in tumor therapy but also in autoimmune diseases, where they may be able to restore immune homeostasis by dampening autoimmune responses and enhancing Treg function. In this study, Kojima et al generate agonistic antibodies against the co-inhibitory receptor TIGIT and a new mouse model that allows for the in vivo assessment of therapeutic that target human TIGIT to be assessed. In a first step, the authors confirm an elevated expression of TIGIT in patients with RA and SLE. They then go on to raise agonistic anti-human TIGIT antibodies which they then functionally test in vivo in the human TIGIT KI mouse model they newly develop. The agonistic anti-TIGIT Ab is able to suppress disease severity in the EAE model and disease parameters in the imiquimod-induced lupus model. They further show that their Ab suppresses Tfh function but enhances Treg-mediated suppression in vitro. Overall, this is an interesting study and the developed mouse model will likely be very helpful to test new tools developed to target human TIGIT. Nevertheless, several aspects of the study should be improved:

Major point:

1. While the mouse model allows for the expression of human TIGIT, the expression levels are strongly reduced. The reasons for and implications of this should be investigated and discussed. Rather than focusing on Tfh, which are rare in naïve mice, (Fig. 2c) the authors should also analyze the expression of ms vs hu TIGIT in Tregs. This should give a clearer impression of the expression levels. Furthermore, it would be important to provide statistics on the expression levels (Suppl. Fig. 3) and perform a proper phenotyping of this newly developed mouse model (T cell development and numbers, lymphocyte populations, etc). How well does mouse CD155 bind to human TIGIT? Does reduced binding/stimulation perhaps lead to reduced TIGIT activity resembling a KO? Does this manifest in changes in disease susceptibility (e.g. the EAE scores seem relatively high)? Also, why did the authors use cDNA (rather than the full human gene) in their construct? Why did they not include the human promotor in the KI, which would have been more representative of the TIGIT expression pattern in human? This should be discussed.

We appreciate the reviewer's instructive suggestions. First, in regards of the levels of KI target gene expression, it is generally known that the expression levels of the KI gene in KI mice are lower than the expression of original gene in wild type mice (Zhu F. et al., Nat. Commun. 2019). For clarification of general knowledge, we cited this article instead of previous #52 (page 21) as follows:

52. Zhu, F., Nair, R. R., Fisher, E. M. C. & Cunningham, T. J. Humanising the mouse genome piece by piece. *Nat. Commun.* **10**, 1–13 (2019).

Currently, it is also not known to manipulate gene expression levels in KI mice. For overcoming lower expression of target gene, generation of transgenic mice expecting higher gene expression can be alternative, but nonspecific overexpression would not be appropriate for our purpose, evaluation of agonistic mAbs. More importantly, TIGIT expression in KI mice increased to sufficient level under inflammatory conditions, whereas low expression in steady state, as noted in Discussion. We believe it is enough to assess function of our mAb in this KI mice. In fact, we could confirm effectiveness of our mAb even in KI mice model. As reviewers' suggestion, we have added this explanation to the 2nd paragraph on page 11 as follows: Thus, we believe it is enough to assess function of our mAb in this KI mice.

We thank the comment on TIGIT expression in Treg and agree with your comments. We already show the mo- and hu-TIGIT expression in separate figures (Fig. 3c and Supplementary Fig. 6), but to improve readability, we have added following sentences to the 2nd paragraph on page 11: Indeed, each hu-TIGIT expression level of Tfh and Treg cells in our hu-TIGIT KI mice was approximately one-fourteenth those of each hu-TIGIT expression level in human cells (Fig. 1a, Fig. 3c, and Supplementary Fig. 6).

We thank the important comment on statics of TIGIT expression in Supplementary Fig. 3 and proper mice immunophenotyping information. We apologize for the lack of these information. We have added statistic information in hu-TIGIT in KI and mo-TIGIT in WT in Supplementary Fig. 4b and the results of T cell immunophenotyping of our generated mice in Supplementary Fig. 3, as follows:

Supplementary Fig. 3 The proportions of Tfh and Tph cells in knock-in (KI) mice are lower than those in wild-type (WT) mice. The proportions of T, B220⁺, CD4⁺ T, CD8⁺ T cells in splenocytes and Tn, Tcm, Tem, Tfh, Tph, Foxp3⁺ Treg cells in CD4⁺ T cells in WT (n=2), HE (n=2) and KI (n=2) 10 weeks female mice are shown. Error bars represent the mean \pm SEM, and *p* values were determined using a two-tailed Student's *t*-test. **p*<0.05.

We have also included the following in the revised results section (the 2nd paragraph in page 5): We evaluated the immunophenotyping in these mice and found the expression of Tfh and Tph cells were lower than those in WT mice (Supplementary Fig. 3). We also evaluated the expression patterns of TIGIT in these mice and found that the Tfh cells in the spleens of KI homozygous or heterozygous mice and wild-type (WT) mice had expression patterns that coincided with the respective mouse genotypes (Supplementary Fig. 4a, b), indicating successful generation of hu-*TIGIT* KI mice.

As these results, the proportions of Tfh and Tph cells were significantly lower in KI compared to WT. In this study, we used only KI mice, our mice developed EAE and IMQ-induced lupus model, and IMQ treatment clearly promoted the production of antibody through increasing Tfh and Tph cells. So, we believed our *in vivo* study could prove the function of our mAb.

In regards of the binding of mo-CD155 to hu-TIGIT in KI mice, it has been reported that mouse CD155 can bind to hu-TIGIT and convey the inhibitory signal (Stanietsky N., et al., Eur. J. Immunol. 2013). Based on the report, we believe it is more likely to wild type rather than TIGIT KO mice in the behaviour of mo-CD155 in our KI mice. Indeed, as reviewer mentioned, it is reported that loss of TIGIT accelerates development of EAE (Joller N., et al., J. Immunol. 2011). We do not have direct proof in our data, but estimated no or quite limited change in disease susceptibility (e.g. the EAE scores) by the effect of mo-CD155 and hu-TIGIT binding themselves in KI mouse.

Finally, in regards of the strategy for generating hu-*TIGIT* KI mice, we selected hu-*TIGIT* cDNA substitution approach. The full length of the hu-*TIGIT* gene is approximately 19 kilo base pairs long so that it is difficult

to clone such long insert to our using vector in efficiency. We also conserved mouse promotor region to use gene expression regulation network in mice, because it has been known for some time that insertion of a human promoter is inhibited by an enhancer (Sanyal A., et al., Nature 2012). As the reviewers suggested, mouse-human difference is so important that we carefully decided to use this approach by the both technical and practical points of view. We strongly believe that we have sufficiently showed the function of our mAb *in vivo*, however we added this concern as limitation in Discussion.

Additional points:

2. In Fig. 4 the authors investigate TIGIT expression upon stimulation. While the change sin frequency of TIGIT expressing cells are limited, it seems like there are large differences in the MFI. It would therefore be important to also analyze the changes in MFI.

We appreciate the reviewer's instructive suggestion. Following this, we also analysed TIGIT and PD-1 change in MFI. In those MFI, PD-1 expression, unlike TIGIT one, was also upregulated in all cells. We have added those results in Fig. 4 and commented in the result section (page 7) as follows:

Mean fluorescence intensity (MFI) also showed that TIGIT was strongly expressed in Tfh cells, while PD-1 was upregulated in all cells (Fig. 4e).

3. It is difficult to understand how the assay to test the biological activity of the Abs (Suppl. Fig. 2) was performed. It would be helpful if the authors included a schematic in the supplementary information.

We thank the reviewer for this important consideration. As you mentioned, it is difficult to understand the assay only text. Then, we have added figures and figure legend to Suppl. Fig. 2 as follows:

Supplementary Fig. 2 The developed mAbs possess agonistic activity against TIGIT-expressing

cells. (a) The schema of the method of agonistic, antagonistic, and cytotoxicity/growth inhibition activity of mAbs are shown. Finally, the luminescence values were measured using Nano-Glo Luciferase Assay System in agonistic and antagonistic activity and CellTiter-Glo Luminescent Cell Viability Assay System in cytotoxicity/growth inhibition activity. We calculated each activity as follows. The agonist activity (%) $= \frac{(1 - ((\text{values of sample}) - (\text{values of 100\% control (PVR } 3.3 \mu\text{g ml}^{-1}))) / ((\text{values of 0\% control (blank)}) - (\text{values of 100\% control (PVR } 3.3 \mu\text{g ml}^{-1})))) \times 100}$. The antagonist activity $= \frac{((\text{values of sample}) - (\text{values of 100\% control (blank)})) / ((\text{values of 0\% control (3.3 } \mu\text{g ml}^{-1} \text{ PVR)}) - (\text{values of 100\% control (blank)})) \times 100}$. The cytotoxicity/growth inhibition activity $= \frac{(1 - ((\text{values of sample}) - (\text{values of 0\% control (blank)})) / ((\text{values of 100\% control (3.3 } \mu\text{g ml}^{-1} \text{ PVR)}) - (\text{values of 0\% control (blank)})) \times 100}$. (b) Agonistic activity, antagonistic activity, and cytotoxicity/growth inhibition activity of isotype control, PVR, and each mAb are shown.

Minor points:

4. It is difficult to understand how the assay to test the biological activity of the Abs (Suppl. Fig. 2) was performed. It would be helpful if the authors included a schematic in the supplementary information.

Please see point 3 above.

5. There is a misleading typo/error on page 6 in the first paragraph, where it read ‘... our mAb significantly suppressed splenomegaly and REDUCED the number of splenocytes...’.

We thank the reviewer for pointing it. The meaning of our sentence was unclear. We wanted to express that the number of splenocytes was increased by IMQ, but our mAb inhibited the increase, so we modified it as follows: our mAb significantly suppressed splenomegaly and THE INCREASE IN the number of splenocytes induced by IMQ treatment.

6. Why did the authors use such a high dose of the Ab in the IMQ model for SLE?

We thank for the meaningful comments. We considered IMQ protocol caused stronger inflammatory condition than EAE one. In our EAE study, we determined the mAb dose from previous studies with anti-mouse TIGIT agonistic antibodies (Dixon, K. O. et al. J. Immunol. 2018). On the other hand, we were the first to use our mAb for the IMQ model. IMQ protocols cause persistent inflammation, and we thought we needed more mAb. We administered the maximum dose available to confirm the function of mAb.

7. The gating strategy for the Foxp3hi Tregs should be included in the supplementary figures.

We appreciate the reviewer's instructive suggestion. I apologize for my lack of explanation. We have added figure legend to Suppl. Fig. 9 as follows: Foxp3^{high} Treg cells were defined as the upper third of the range containing 98% of Foxp3⁺ Treg cells in CD4⁺ T cells.

8. The last sentence of the second paragraph on page 7 should read most pronounced rather than most selective as the effect is not exclusive.

We agree with your assessment and have changed “selective” into “pronounced”.

9. On page 7 the authors state that there is a stronger inhibitory effect of the anti-TIGIT Ab on Tfh than on non-Tfh. While this is true, it would be important to mention that the non-Tfh cells are also strongly suppressed.

Thank you for your suggestion and apologize for confusing. Our mAb worked TIGIT-expressing cells and the function was dependent on the levels of TIGIT expression. It is very important point for our mAb to suppress not only Tfh cells but also non-Tfh cells. Then, we have modified the sentence (page 7) as follows: There was no difference in cell viability (Fig. 5a, c), and although both cells were inhibited cell proliferation, a stronger inhibitory effect on cell proliferation was observed for the Tfh cells than the non-Tfh cells (Fig. 5b, d).

Reviewer #2 (Remarks to the Author):

Communications Biology manuscript TITLE: “Anti-human-TIGIT agonistic antibody ameliorate autoimmune diseases by restoring T cell imbalance” presented interesting findings. The strengths of the manuscript included generation of anti-human -TIGIT agonistic monoclonal antibodies and human TIGIT knock-in mice, and findings of the mABs’ advantages in inhibiting autoimmune T cell responses and enhancing immunosuppressive Treg function. However, the manuscript needs to be improved by fixing the following major concerns in presentation and analysis of the results.

There are major concerns:

1. The last part of manuscript title is misleading. The last part of the title can be revised into “...by inhibiting autoimmune Tfh and Tph cells and enhancing Tregs”;

We appreciate the reviewer’s instructive suggestion. We revised our title into “Anti-human-TIGIT agonistic antibody ameliorates autoimmune diseases by inhibiting Tfh and Tph cells and enhancing Treg cells”

2. The authors did not present any significant findings from human TIGIT knock-in mice in the Abstract and the Results section;

Thank you for providing these insights. We agree that hu-TIGIT KI mice are also important and have added the results of immunophenotyping of them in Supplementary Fig. 3 (reviewer #1 major point). In addition, the importance of using KI mice was that we proved the function of our same mAb both *in vitro* and *in vivo*. We considered that most significant findings from hu-TIGIT KI mice is to prove the function of our mAb and wrote the results in the Abstract.

3. The authors need to use summary sentences as the titles for all the figures;

We apologize our insufficient titles for all figures and have changed them to summary sentences as follows:

Fig. 1 TIGIT-expressing CD4⁺ T cell subsets correlate disease activity in patients with systemic autoimmune diseases.

Fig. 2 Anti-human-TIGIT agonistic mAb (α TIGIT) ameliorates EAE human-TIGIT (hu-TIGIT) knock-in (KI) mice.

Fig. 3 Anti-human-TIGIT (anti-hu-TIGIT) agonistic mAb (α TIGIT) suppresses T cell dependent autoantibody production in a hu-TIGIT knock-in (KI) imiquimod (IMQ)-induced lupus model in mice.

Fig. 4 Activation induced TIGIT expression is specific in Tfh cells.

Fig. 5 Anti-human-TIGIT agonistic mAb (α TIGIT) inhibits B cell activation via Tfh suppression.

Fig. 6 Anti-hu-TIGIT agonistic mAb (α TIGIT) enhances suppressive activity of naïve Treg cells.

Fig. 7 Anti-hu-TIGIT agonistic mAb (α TIGIT) regulates T cell imbalance in autoimmune states.

Supplementary Fig. 1 Some population of effector T cells increased in systemic autoimmune diseases.

Supplementary Fig. 2 The developed mAbs possess agonistic activity against TIGIT-expressing cells.

Supplementary Fig. 3 The proportions of Tfh and Tph cells in knock-in (KI) mice are lower than those in wild-type (WT) mice.

Supplementary Fig. 4 Human- (Hu-) and mouse- (mo-)TIGIT expression coincides with the respective mouse genotypes.

Supplementary Fig. 5 Anti-human-TIGIT (anti-hu-TIGIT) agonistic mAb (α TIGIT) also suppresses the proliferation of Tfh and Tph cells of lymphocytes in a hu-*TIGIT* knock-in (KI) imiquimod (IMQ)-induced lupus model in mice.

Supplementary Fig. 6 TIGIT expression of Treg cells is upregulated in patients with RA compared with HCs.

4. All the figures need to be self-telling so that reviewers and readers understand the figures right away, rather than letting them looking around and trying to understand the figures.

Thank you for your very important suggestion. As you mentioned, our Figures were difficult to see because they were moving back and forth, so we have arranged Fig. 2, Fig. 3, and Supplementary Fig. 4 as following:

Fig. 2 Anti-human-TIGIT agonistic mAb (α TIGIT) ameliorates EAE human-*TIGIT* (hu-*TIGIT*) knock-in (KI) mice. **a** The sorted Tfh cells were labelled with CellTrace violet (CTV) and activated with anti-CD3 and anti-CD28 Abs with plate-bound α TIGIT (or the appropriate isotype control) for 4 days. CTV dilution was analysed by flow cytometry. The suppression rate was calculated as follows: $(1 - (\text{proliferation rate of mAb} / (\text{proliferation rate of isotype}))) \times 100$ (mean + SEM) is shown (n=2). **b** The strategy for generating hu-*TIGIT* KI mice is shown. **c** EAE immunisation and α TIGIT treatment protocols are shown. **d** EAE mice were

divided into treated (n=6, red) and nontreated (n=6, black) groups, and the mean clinical score + or - SEM is shown. *P* values were determined by a two-tailed Student's *t*-test. **p*<0.05.

Fig. 3 Anti-human-TIGIT (anti-hu-TIGIT) agonistic mAb (αTIGIT) suppresses T cell dependent autoantibody production in a hu-TIGIT knock-in (KI) imiquimod (IMQ)-induced lupus model in mice.

a The method used to establish IMQ-induced lupus model mice and the αTIGIT treatment protocol are shown. **b** Hu-TIGIT KI mice were divided into three groups: isotype-treated (n=4, black open circle), IMQ-

and α TIGIT-treated (n=5, red closed circle), and IMQ- and isotype-treated (n=5, black closed circle) mice. Spleen weight/body weight and total splenocyte number are shown. **c** TIGIT expression of various CD4⁺ T cell subsets and that of B220⁺ cells are shown. **d** The proportion of CD4⁺ T cells in splenocytes and Tn, Tcm, and Tem in CD4⁺ T cells are shown. **e** Representative flow cytometry plots for Tph cells and Tfh cells and the proportions of those cells are shown. **f** The proportion of Foxp3⁺ Treg cells among CD4⁺ T cells and Foxp3^{high} Treg cells (**g**) among Foxp3⁺ Treg cells are shown. **h** The proportions of B220⁺ cells in the spleen, **i** Germinal centre (GC) B cells among B220⁺ cells and **j** plasma cells in total splenocytes are shown. **k** Anti-dsDNA Abs in plasma at 6 weeks are shown (one sample could not be collected in IMQ- and isotype-treated). Error bars represent the mean \pm SEM, and *p* values were determined using a two-tailed Student's *t*-test. **p*<0.05, ***p*<0.01, ****p*<0.001.

Supplementary Fig. 4 Human- (Hu-) and mouse- (mo-)TIGIT expression coincides with the respective mouse genotypes. Hu-TIGIT and mo-TIGIT expression of whole CD4⁺ T, Tfh, Treg, and whole CD8⁺ T cells in WT (n=2), HE (n=2) and KI (n=2) female mice are checked, and representative flow cytometry plots of Tfh cells (**a**) and each expression (**b**) are shown. Error bars represent the mean \pm SEM, and *p* values were determined using a two-tailed Student's *t*-test. **p*<0.05. In regards of (**b**), we compared the proportion of mo-TIGIT expression in WT with hu-TIGIT expression in KI.

Revising the manuscript stated above, the original figure numbers are changed from Fig. 2c to Supplementary Fig. 3a, from Fig. 2d to Fig. 2c, from Fig. 2e to Fig. 2d, from Fig. 3c to Fig. 3d, from Fig. 3d to Fig. 3e, from Fig. 3e to Fig. 3f, from Fig. 3f to Fig. 3g, from Fig. 3g to Fig. 3h, from Fig. 3h to Fig. 3i, from Fig. 3i to Fig. 3j, from Fig. 3j to Fig. 3k, from Supplementary Fig. 2 to Supplementary Fig. 2b and from Supplementary Fig. 3 to Supplementary Fig. 4b.

Old version	New version
2c	Sup. 3a
2d	2c
2e	2d

3c	3d
3d	3e
3e	3f
3f	3g
3g	3h
3h	3i
3i	3j
3j	3k
	Sup. 2a
Sup. 2	Sup. 2b
	Sup. 3
Sup. 3	Sup. 4b

Reviewers' comments:

Reviewer #2 (Remarks to the Author):

Unfortunately, the issue I had raised were not really addressed by the authors. My major concern was related to the reduced expression levels of TIGIT in the humanized mouse model and the poor characterization of the model. The authors reply that 'in regards of the levels of KI target gene expression, it is generally known that the expression levels of the KI gene in KI mice are lower than the expression of original gene in wild type mice (Zhu F. et al., Nat. Commun. 2019).' However, this review article actually stated that 'humanised gene expression levels in mouse closely correlated with levels of the mouse orthologous gene' but not necessarily the human gene. Furthermore, it concludes that expression levels can be significantly lower if larger constructs are inserted, however in the current study that is surely not the issue given that cDNA was used for the construct. Furthermore, reference to higher expression levels in inflammatory settings is also not satisfactory as expression of endogenous mouse TIGIT would also be expected to be highly increased. With expression levels that are <10% of the natural expression the statement that this allows for proper assessment of its function in disease is a bold statement.

Furthermore, functionality of the interaction of mouse CD155 with the huTIGIT expressed in this model was not tested but just inferred. This needs to be shown experimentally in this setting.

Reviewer #3 (Remarks to the Author):

No additional concerns

Reviewers' comments:

Reviewer #2 (Remarks to the Author):

Unfortunately, the issue I had raised were not really addressed by the authors. My major concern was related to the reduced expression levels of TIGIT in the humanized mouse model and the poor characterization of the model. The authors reply that 'in regards of the levels of KI target gene expression, it is generally known that the expression levels of the KI gene in KI mice are lower than the expression of original gene in wild type mice (Zhu F. et al., Nat. Commun. 2019).' However, this review article actually stated that 'humanised gene expression levels in mouse closely correlated with levels of the mouse orthologous gene' but not necessarily the human gene. Furthermore, it concludes that expression levels can be significantly lower if larger constructs are inserted, however in the current study that is surely not the issue given that cDNA was used for the construct. Furthermore, reference to higher expression levels in inflammatory settings is also not satisfactory as expression of endogenous mouse TIGIT would also be expected to be highly increased. With expression levels that are <10% of the natural expression the statement that this allows for proper assessment of its function in disease is a bold statement. Furthermore, functionality of the interaction of mouse CD155 with the huTIGIT expressed in this model was not tested but just inferred. This needs to be shown experimentally in this setting.

Reviewer #3 (Remarks to the Author):

No additional concerns

29th March, 2023

Dear Reviewers,

Thank the editor and reviewers for giving us opportunity to revise our manuscript (COMMSBIO-22-0736) titled "Anti-human-TIGIT agonistic antibody ameliorates autoimmune diseases by inhibiting Tfh and Tph cells and enhancing Treg cells". We sincerely appreciate your decision and intensively revised it as your instruction as possible as we can.

Reviewers' comments:

Reviewer #2 (Remarks to the Author):

Unfortunately, the issue I had raised were not really addressed by the authors. My major concern was related to the reduced expression levels of TIGIT in the humanized mouse model and the poor characterization of the model. The authors reply that 'in regards of the levels of KI target gene expression, it is generally known that the expression levels of the KI gene in KI mice are lower than the expression of original gene in wild type mice (Zhu F. et al., Nat. Commun. 2019).' However, this review article actually stated that 'humanised gene expression levels in mouse closely correlated with levels of the mouse orthologous gene' but not necessarily the human gene. Furthermore, it concludes that expression levels can be significantly lower if larger constructs are inserted, however in the current study that is surely not the issue given that cDNA was used for the construct. Furthermore, reference to higher expression levels in inflammatory settings is also not satisfactory as expression of endogenous mouse TIGIT would also be expected to be highly increased. With expression levels that are <10% of the natural expression the statement that this allows for proper assessment of its function in disease is a bold statement. Furthermore, functionality of the interaction of mouse CD155 with the huTIGIT expressed in this model was not tested but just inferred. This needs to be shown experimentally in this setting.

Thank you very much for your valuable comments on our manuscript. We apologize for our misunderstanding and misrepresentation regarding the KI gene you pointed out. It is unknown whether humanized mice have lower gene expression. We back down on the citation #52 we added last time.

To answer reviewers' questions, we confirmed that hu-TIGIT expression in KI mice increased to the same level as mo-TIGIT expression in WT mice under inflammatory conditions, and that mo-TIGIT expression in KI mice was not increased even under inflammatory conditions. In addition, we showed our mAb did not recognise mo-TIGIT. We have added the following to page 6, Sup. 5, page5, and Sup. 2c and revised the description of page 11 as follows.

Page6

In IMQ-induced lupus model, the proportion of hu-TIGIT-expressing cells in KI mice was comparable to that of mo-TIGIT-expressing cells in WT mice, and mo-TIGIT-expressing cells in KI mice were not increased (Supplementary Fig. 5a, b).

Supplementary Fig. 5 In the imiquimod (IMQ)-induced lupus model, elevated human TIGIT (hu-TIGIT) expression in knock-in (KI) mice is similar to elevated mouse TIGIT (mo-TIGIT) in wild-type (WT) mice. Hu- and mo-TIGIT expression in IMQ-induced lupus model in WT mice (n=2) and those in KI mice (n=2) of whole CD4⁺ T, Tfh and Tph cells are checked, and representative flow cytometry plots of Tfh cells (a) and each expression (b) are shown. Error bars represent the mean ± SEM, and *p* values were determined using a two-tailed Student's *t*-test. **p*<0.05. We compared the proportion of mo-TIGIT expression in WT with hu-TIGIT expression in KI.

Page11

Our use of KI mice enabled us to demonstrate the effect of each mAb type both *in vitro* and *in vivo*, but this aspect of the study was associated with a certain limitation: The hu-TIGIT expression level of Tfh cells in our KI mice was approximately one-eighth that of the mouse-TIGIT expression level in WT mice and one-twentieth that of the hu-TIGIT expression level in human cells (Fig. 1a and Fig. 3c). Nevertheless, we were able to confirm that the effects of our mAb on T cells were the same *in vitro* and *in vivo* and to demonstrate the efficacy of this mAb in hu-TIGIT KI mice. We consider this is because hu-TIGIT expression in KI mice increased to the same level as mo-TIGIT in WT mice under inflammatory conditions (Supplementary Fig. 5a, b). Thus, we believe it is enough to assess function of our mAb in this KI mice. Furthermore, this low expression in hu-TIGIT KI mice suggests that our mAb may be even more effective in humans, as humans express higher levels of TIGIT.

Page5

We also evaluated its cross-recognition, and M1-8 recognise hu- and macaque-TIGIT, but not mo-TIGIT (Supplementary Fig. 2c).

c Cross-recognition of M1-8 and human, macaque, and mouse-TIGIT is shown.

It has been reported that mouse CD155 can bind to hu-TIGIT and convey the inhibitory signal (Stanietsky N., et al., Eur. J. Immunol. 2013). We do not directly assess interaction of mouse CD155 with hu-TIGIT in our system. However, the untreated group was used as controls, and even if mouse CD155 effects, that effect would not affect the assessment of efficacy of our mAbs *in vivo*. We have added the following to page 11 as a limitation.

There is another limitation of our study. It has been reported that mouse CD155 can bind to hu-TIGIT and convey the inhibitory signal⁵². We do not directly assess this binding *in vivo*. However, even if the mouse CD155 interacts hu-TIGIT, we believe that our experiments were able to prove the effect of the mAb itself by using an untreated group as a control.

Finally, we compared T cell development and spleen weights of WT, HE, and KI mice as a characterization. We have added to page5, Sup. 3 as following.

Page5

We evaluated the immunophenotyping of thymocytes and splenocytes in these mice. There was no significant difference in T cell development (Supplementary Fig. 3a).

a Flow cytometric analysis of T cell development in WT (n=2), heterozygous (HE) (n=2) and KI (n=2) mice is checked. Representative flow cytometry plots of thymocytes and the proportions of CD4+, CD8+, and double positive (DP) T cells are shown. **b** The spleen weight in WT (n=2), HE (n=2) and KI (n=2) 10 weeks female mice is shown.

Revising the manuscript stated above, the original figure numbers are changed from Supplementary Fig. 3 to Supplementary Fig. 3b, from Supplementary Fig. 5 to Supplementary Fig. 6, from Supplementary Fig. 6 to Supplementary Fig. 7, from Supplementary Fig. 7 to Supplementary Fig. 8, from Supplementary Fig. 8 to Supplementary Fig. 9, and from Supplementary Fig. 9 to Supplementary Fig. 10. We also added new version to Supplementary Fig. 2c, Supplementary Fig. 3a, Supplementary Fig. 5, and Supplementary Fig. 11.

Old version	New version
	Sup. 2c
	Sup. 3a
Sup. 3	Sup. 3b

	Sup. 5
Sup. 5	Sup. 6
Sup. 6	Sup. 7
Sup. 7	Sup. 8
Sup. 8	Sup. 9
Sup. 9	Sup. 10
	Sup. 11

REVIEWERS' COMMENTS:

Reviewer #2 (Remarks to the Author):

The authors have sufficiently addressed my remaining questions.